# Midbrain signaling of identity prediction errors depends on orbitofrontal cortex networks

Published online: 24 Februrary 2024
Qingfang Liu [1], Yao Zhao[1], Sumedha Attanti [2], Joel L. Voss[3], Geoffrey Schoenbaum [1] & Thorsten Kahnt [1] ✉

Outcome-guided behavior requires knowledge about the identity of future rewards. Previous work across species has shown that the dopaminergic midbrain responds to violations in expected reward identity and that the lateral orbitofrontal cortex (OFC) represents reward identity expectations. Here we used network-targeted transcranial magnetic stimulation (TMS) and functional magnetic resonance imaging (fMRI) during a trans-reinforcer reversal learning task to test the hypothesis that outcome expectations in the lateral OFC contribute to the computation of identity prediction errors (iPE) in the midbrain. Network-targeted TMS aiming at lateral OFC reduced the global connectedness of the lateral OFC and impaired reward identity learning in the first block of trials. Critically, TMS disrupted neural representations of expected reward identity in the OFC and modulated iPE responses in the midbrain. These results support the idea that iPE signals in the dopaminergic midbrain are computed based on outcome expectations represented in the lateral OFC.

Knowledge about associations between cues and outcomes is fundamental for adaptive behavior. This involves not only the value of expected outcomes, but also the value-neutral sensory characteristics that comprise their identity. Value-neutral representations of expected outcomes are part of a detailed model of the world that can be used to support flexible model-based planning and decision making in changing environments (see[1–4] for reviews).

Recent work suggests that the dopaminergic midbrain contributes to reward identity learning. Specifically, prediction error (PE) signaling in midbrain dopamine neurons[5–8] is not restricted to mismatches in value predictions but can also be observed when expectations about value-neutral reward features are violated[9–16]. For instance, the same dopaminergic neurons in rats respond to unexpected changes in both the magnitude (i.e., value) and flavor (i.e., identity) of equally-preferred food rewards[9,16]. Similarly, functional magnetic resonance imaging (fMRI) responses in overlapping regions of the human midbrain correlate with identity (iPE) and value PEs[10–12].

Moreover, activity of dopaminergic neurons at the time of reward is critical for establishing outcome-specific associations between sensory cues and rewards[17]. These findings suggest a key role for dopaminergic iPEs in reward identity learning, but the systems-level mechanisms by which these error signals are computed remain unclear.

A candidate region for providing detailed information about the identity of expected outcomes as input for computing iPEs is the lateral orbitofrontal cortex (OFC)[18]. Work across species indicates that lateral OFC maintains information about the identity of expected outcomes[2,3,10,19–24]. For instance, neural responses in the lateral OFC correlate with the identity of future rewards and track changes in outcome identity across reversals[10,19,21,25]. Moreover, lesions or inactivation of the lateral OFC cause deficits in behaviors that require information about the identity of expected outcomes[26–29].

Given the involvement of the lateral OFC and midbrain in signaling reward identity and reward identity errors, respectively, as well as

[1]National Institute on Drug Abuse Intramural Research Program, Baltimore, MD 21224, USA. [2]Mayo Clinic Alix School of Medicine, Scottsdale, AZ 85259, USA. [3]Department of Neurology, The University of Chicago, Chicago, IL 60611, USA. ✉e-mail: thorsten.kahnt@nih.gov

their positioning within the cortico-striatal circuit[30,31], we hypothesized that both regions jointly contribute to reward identity learning. Specifically, we hypothesized that reward identity expectations represented in the lateral OFC might contribute to the computation of iPEs signaled by the midbrain, which, in turn, update reward identity expectations in the lateral OFC.

A key prediction from this hypothesis is that disrupting activity in the lateral OFC should alter midbrain responses to iPEs. Here, we causally tested this prediction in humans. To this end, we used network-targeted transcranial magnetic stimulation (TMS) to disrupt activity in the lateral OFC network[26,32] and used fMRI to measure midbrain responses to iPEs in a trans-reinforcer reversal learning task. Similar to tasks that we have previously used to demonstrate iPE signaling in the rat and human midbrain[9–11,21], the task required subjects to learn associations between visual cues and unique but equally-preferred food odor rewards. These associations reversed unpredictably throughout the task, thereby inducing iPEs. We used a computational model of reward identity learning to derive predictions for TMS-induced changes in task behavior, midbrain iPE signals, and OFC identity expectations. In line with the predictions of a model with reduced learning rates, we found that OFC network-targeted TMS (1) impairs reward identity learning, (2) modulates iPE signaling in the midbrain, and (3) disrupts reward identity expectations in the OFC.

These results support the idea that reward identity learning depends on recursive interactions between outcome expectations in the lateral OFC and error signals in the dopaminergic midbrain.

## Results

### Trans-reinforcer reversal learning task and experimental design

We studied identity learning using a trans-reinforcer reversal learning task (Fig. 1a). On each trial, subjects (N = 31, 11 male, age 19–42) saw one of two visual cues that were deterministically paired with one of three unique but equally-valued food odor rewards (e.g., potato chips, chocolate, and peach, Fig. 1d). During cue presentation, subjects were asked to predict which odor would be delivered after the cue. Thus, subjects needed to learn and maintain the associations between the cues and the food odors. Unpredictably to the subject, the associations between cues and odors reversed six times for each cue per task run, thereby eliciting iPEs (Fig. 1b).

The study was conducted over 4 days (Fig. 1e): an initial screening session (Day 1), a session to obtain a structural MRI, a resting-state fMRI scan, and to determine resting motor thresholds (rMT) (Day 2), and two TMS sessions (Day 3 and Day 4) wherein cTBS targeting the lateral OFC network (or sham, both outside the MRI scanner) was followed by three runs of the trans-reinforcer reversal learning task inside the MRI scanner.

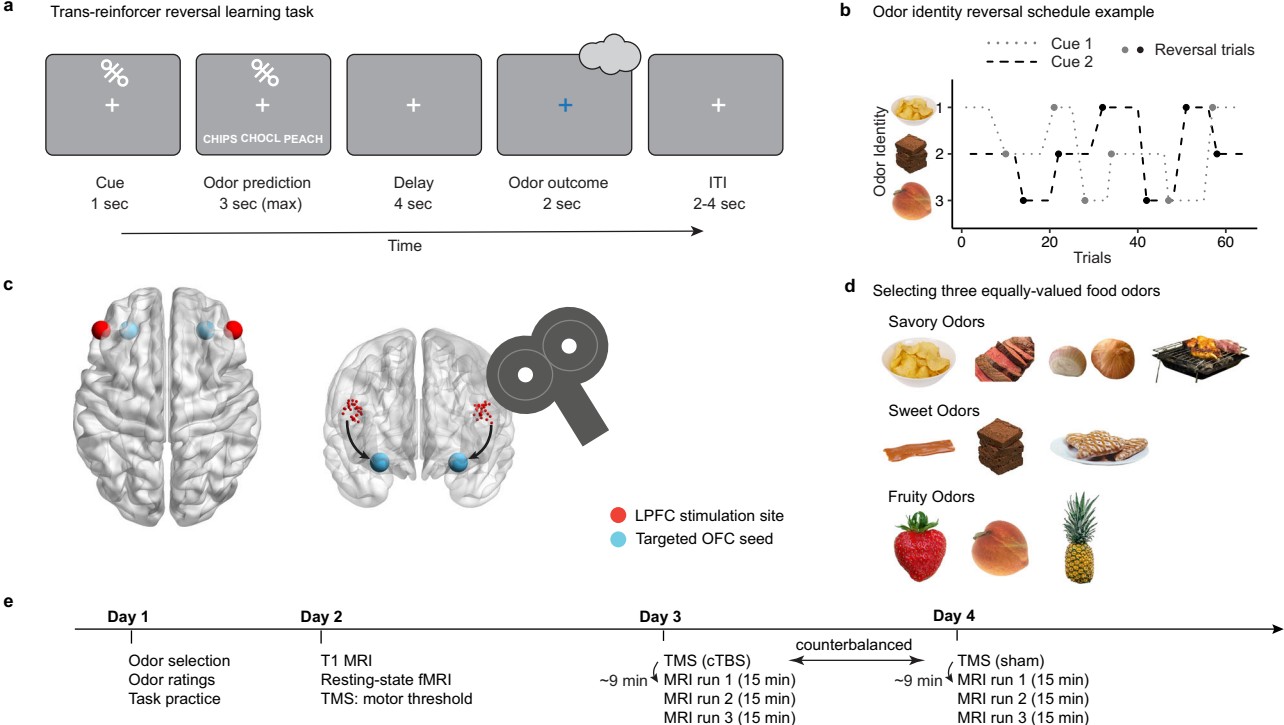

**Fig. 1 | Trans-reinforcer reversal learning task and experimental design.**
**a** Schematic of the trans-reinforcer reversal learning task. Each trial starts with the presentation of a visual cue (1 s), followed by a prompt to predict which odor is currently associated with the presented cue (3 s). After 4 s of fixation, an odor is delivered (2 s), with the fixation turning blue cueing the subject to sniff. Each trial ends with an inter-trial interval pseudorandomly drawn from 2 to 4 s. **b** Odor identity reversal schedule example. Trials with the two visual cues are randomly interleaved. On a given trial, each visual cue is associated with one of three equally-valued odors, and the association is reversed after a variable number of trials. Trials in which the associated odor changes from the previous trial with the same cue are labeled "reversal trials." **c** Illustration of network-targeted TMS on a glass brain using BrainNet viewer[65]. left, indirectly targeted OFC seed region (cyan) and the LPFC stimulation site (red); right, individually selected stimulation coordinates in the LPFC (red) based on maximal resting-state fMRI connectivity with the targeted

OFC seed region (cyan). **d** Three different categories of food odors were used in the experiment: savory, sweet, and fruity. For each subject, one odor was selected from each category such that they matched in pleasantness ratings (as a proxy for reward value) collected on Day 1. **e** Study timeline. Day 1: subjects rated the pleasantness and intensity of odors to select the three equally-valued odors. Day 2: T1-weighted brain image and resting-state fMRI scans were collected, followed by determination of the resting motor threshold (rMT). Day 3/4: TMS (cTBS or sham, order counterbalanced across subjects) session, followed by three runs of task-based fMRI scans. The average time from finishing TMS to the start of the MRI scan was ~9 min. Each fMRI run took approximately 15 min. ITI inter-trial interval, LPFC lateral prefrontal cortex, OFC orbitofrontal cortex, TMS transcranial magnetic stimulation, MRI magnetic resonance imaging, fMRI functional magnetic resonance imaging, cTBS continuous theta-burst stimulation.

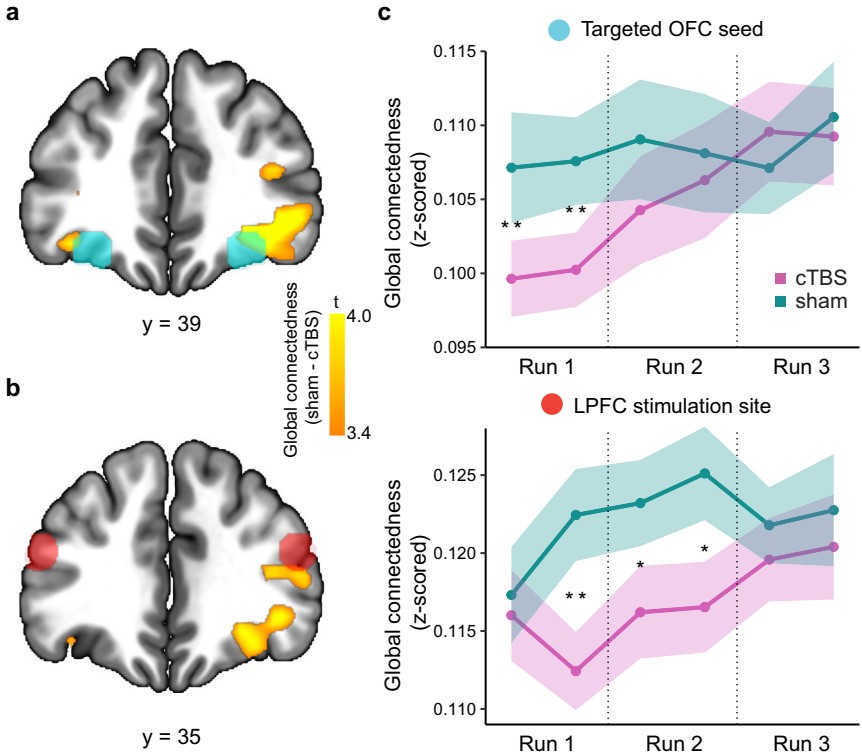

**Fig. 2 | Network-targeted TMS decreases global connectedness in lateral OFC.**
**a** Difference in global connectedness between sham and cTBS in the first run, overlaid with the indirectly targeted OFC seed regions (cyan spheres). Statistical map depicts t-values for the difference in global connectedness between the cTBS and sham session, thresholded at $p < 0.001$ (uncorrected) for illustration. Right OFC: [33, 38, −10], t(30) = 3.97, $p_{FWE-SVC}$ = 0.002; left OFC: [-33, 38, −13], t(30) = 3.50, $p_{FWE-SVC}$ = 0.022. **b** Same as (**a**), but the red spheres indicate the LPFC stimulation sites. Right LPFC: [45, 35, 14], t(30) = 3.65, $p_{FWE-SVC}$ = 0.014. **c** Changes in global connectedness across time in each session (cTBS, sham) and ROI (targeted OFC seed, LPFC stimulation site). Global connectedness values were calculated by splitting each run into half (6 time bins in total). The line plots denote the mean

global connectedness, and the shaded areas depict ±1 standard error across subjects. Differences between sessions were tested at each time bin using Wilcoxon signed rank test (V, one-sided). OFC: V = 114, $p$ = 0.0038 (1st time bin); V = 128, $p$ = 0.0088 (2nd time bin). LPFC: V = 118, $p$ = 0.0049 (2nd time bin); V = 150, $p$ = 0.028 (3rd time bin); V = 143, $p$ = 0.020 (4th time bin). * denotes $p < 0.05$, and ** denotes $p < 0.01$. See Supplementary Fig 2 using the functional ROIs shown in (**a**) and (**b**) for subject-wise TMS effects on global connectedness. Correcting for head motion does not alter these results (see Supplementary Fig 4). LPFC lateral prefrontal cortex, OFC orbitofrontal cortex, cTBS continuous theta-burst stimulation, FWE-SVC family-wise error small-volume correction. Source data are provided as a Source Data file.

Our TMS approach targeted an area in the lateral OFC, which has previously been shown to represent the identity of expected rewards[19,22,33] and which is functionally connected to an isolated cluster in the lateral prefrontal cortex (LPFC) in a large normative data set (neurosynth.org). For each subject and hemisphere, we then used this lateral OFC area as the seed region in a subject-level resting-state fMRI connectivity analysis and identified an area in LPFC that showed maximal connectivity with the OFC seed (Fig. 1c). Our previous work has shown that continuous theta burst stimulation (cTBS) over these individually selected stimulation sites modulates activity in the lateral OFC network and disrupts outcome-guided behaviors[26,32].

Each subject received both cTBS and sham in different sessions, with the order counterbalanced across subjects. This design enabled us to examine the effect of cTBS on reward identity learning and neural responses to both iPEs and reward identity expectations within each subject, and thereby to investigate the causal relationship between OFC network activity and midbrain error signaling during reward identity learning.

### Network-targeted TMS decreases global connectedness in lateral OFC

In a first step, we verified the effects of OFC-targeted cTBS on activity in the lateral OFC, as in our previous study[26]. Specifically, we calculated a voxel-wise measure of global connectedness[26] using the fMRI data from the first run of each session. Global connectedness was

computed as the average absolute correlation between each voxel's fMRI time series and that of every other gray matter voxel in the brain. Comparing sham and cTBS sessions, we found that cTBS decreased the global connectedness in the indirectly targeted OFC areas (right OFC: [33, 38, −10], t(30) = 3.97, $p_{FWE-SVC}$ = 0.002; left OFC: [−33, 38, −13], t(30) = 3.50, $p_{FWE-SVC}$ = 0.022, Fig. 2a). We found a similar decrease in connectedness in the actual stimulation site in the right LPFC ([45, 35, 14], t(30) = 3.65, $p_{FWE-SVC}$ = 0.014, Fig. 2b). These results show that network-targeted TMS disrupted activity in the lateral OFC network, as intended.

To track the effects of cTBS on connectivity across time, we divided the fMRI time series into six time bins (i.e., two bins per run) and computed the global connectedness for each time bin in independent ROIs of the OFC and LPFC (Fig. 2c). Compared to sham, cTBS decreased the global connectedness of the OFC in the first (V = 114, $p$ = 0.0038) and second time bin (V = 128, $p$ = 0.0088), but not in any other bin (all p's > 0.1). Importantly, there was a significant interaction between time and session in the OFC ($p$ = 0.017) that was driven by global connectedness returning to baseline after cTBS ($p$ = 3.73e−8), and no change after sham ($p$ = 0.41). Global connectedness of the LPFC was decreased in the second (V = 118, $p$ = 0.0049), third (V = 150, $p$ = 0.028), and fourth (V = 143, $p$ = 0.020) time bin, but not in any other bin (all p's > 0.1). Although global connectedness in the LPFC returned to baseline after cTBS ($p$ = 0.0067) and there was no significant change after sham stimulation ($p$ = 0.12), the time by session interaction was not significant ($p$ = 0.53).

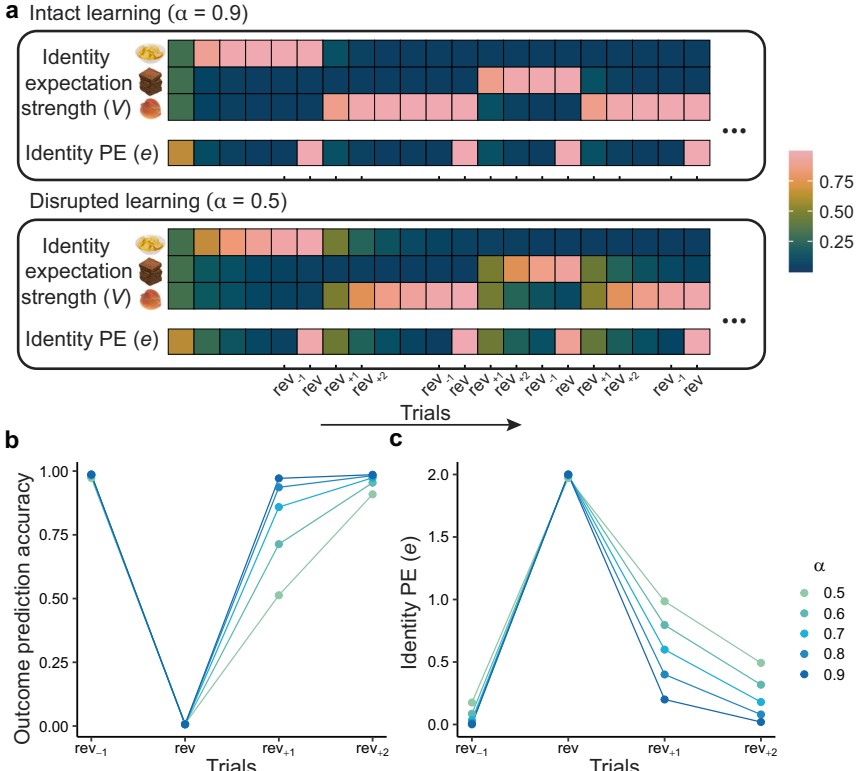

**Fig. 3 | Modeling cTBS-induced impairments in reward identity learning.**
**a** Simulated across-trial updates of identity expectation strengths (*V*) and identity prediction errors (*e*) based on two learning rates: $\alpha = 0.9$ (intact learning, i.e., sham) and $\alpha = 0.5$ (disrupted learning, i.e., cTBS). The identity expectation strengths corresponding to three odor identities (chips, chocolate, and peach, as an example) and the same cue evolve across trials, in response to odor identity reversals and a delta learning rule. The identity PEs (discrepancy between the received reward identity and the expectation strengths across the three reward identities) are normalized within each learning rate simulation, such that the lowest and highest identity PEs are equal to 0 and 1, respectively. **b** Simulated outcome prediction accuracy for $rev_{-1}$, $rev$, $rev_{+1}$, and $rev_{+2}$ trials, under learning rates $\alpha$ from 0.5 to 0.9. **c** Simulated identity prediction errors (*e*) for $rev_{-1}$, $rev$, $rev_{+1}$, and $rev_{+2}$ trials, under learning rates $\alpha$ from 0.5 to 0.9. rev: reversal; PE prediction error.

We also tested whether OFC-targeted cTBS modulated global connectedness in additional brain areas. First, we ran a whole brain analysis ($p_{FWE} < 0.05$, whole-brain FWE corrected), which did not reveal any significant effects of TMS. Second, we explored effects of TMS on connectedness in several other brain areas commonly associated with reward learning (Supplementary Fig 3).

**Modeling cTBS-induced impairments in reward identity learning**
To generate predictions for our behavioral and neural data, we designed an identity learning model in which identity expectations are updated by iPEs (Fig. 3a), computed as the difference between expected and received odor outcomes[11,26]. We hypothesized that modulating lateral OFC network function through cTBS would disrupt reward identity expectations. To implement this computationally, we reduced the rates at which reward identity expectations were updated during learning. We performed simulations with varying learning rates (ranging from 0.5 to 0.9) to generate predictions for the possible effects of cTBS on behavioral and neural data (Fig. 3). With a high learning rate ($\alpha = 0.9$), the expectation strengths are rapidly updated to the new odor identity after each reversal, leading to intact reward identity expectations and identity PEs. Conversely, with a low learning rate ($\alpha = 0.5$), the expectation strengths are slower to update to the new odor identity after reversals, causing reduced identity expectations and prolonged identity PEs. Taken together, these simulations show that the effects of different learning rates on both outcome prediction accuracy and iPEs are most evident on $rev_{+1}$ trials (Fig. 3b, c).

**cTBS disrupts outcome prediction accuracy**
We next leveraged the results of the model simulation shown in Fig. 3b to predict the effects of cTBS on subjects' behavior in the task. Collapsing data from all three experimental runs and both sessions, accuracy significantly decreased from $rev_{-1}$ to $rev$ trials (paired *t* test, t(30) = 27.46, $p < 2.2e{-}16$, two-sided) and recovered from $rev$ trials to $rev_{+1}$ trials (paired *t* test, t(30) = 27.76, $p < 2.2e{-}16$, two-sided). In addition, there was a decrease in accuracy from $rev_{-1}$ to $rev_{+1}$ trials (paired *t* test, t(30) = 3.88, $p = 5.4e{-}4$, two-sided), suggesting that performance was lower directly after compared to directly before reversals. Importantly, our model predicted that this decrease from before to after the reversal would be more pronounced with lower learning rates (Fig. 3b). We therefore next examined the effect of cTBS on the change in accuracy from $rev_{-1}$ to $rev_{+1}$ trials.

Given the transient effects of cTBS on global connectedness (Fig. 2c), we reasoned that any effect of cTBS on behavior should be most evident in the first run. Indeed, we found that in the first run, accuracy decreased significantly from $rev_{-1}$ to $rev_{+1}$ trials in the cTBS session (paired *t* test, t(30) = 3.013, $p = 0.0026$, one-sided), but not in the sham session (paired *t* test, t(30) = 0.319, $p = 0.62$, one-sided). Most importantly, this change in accuracy from $rev_{-1}$ to $rev_{+1}$ trials was significantly different between the cTBS and sham session (Fig. 4a, b, paired *t* test, t(30) = 2.344, $p = 0.013$, one-sided). No such effect of TMS on the change in accuracy was found in either the second or the third run (both $p > 0.099$), consistent with a recovery of function as the effects of TMS waned. Averaging across three runs, there was no significant effect of cTBS on the

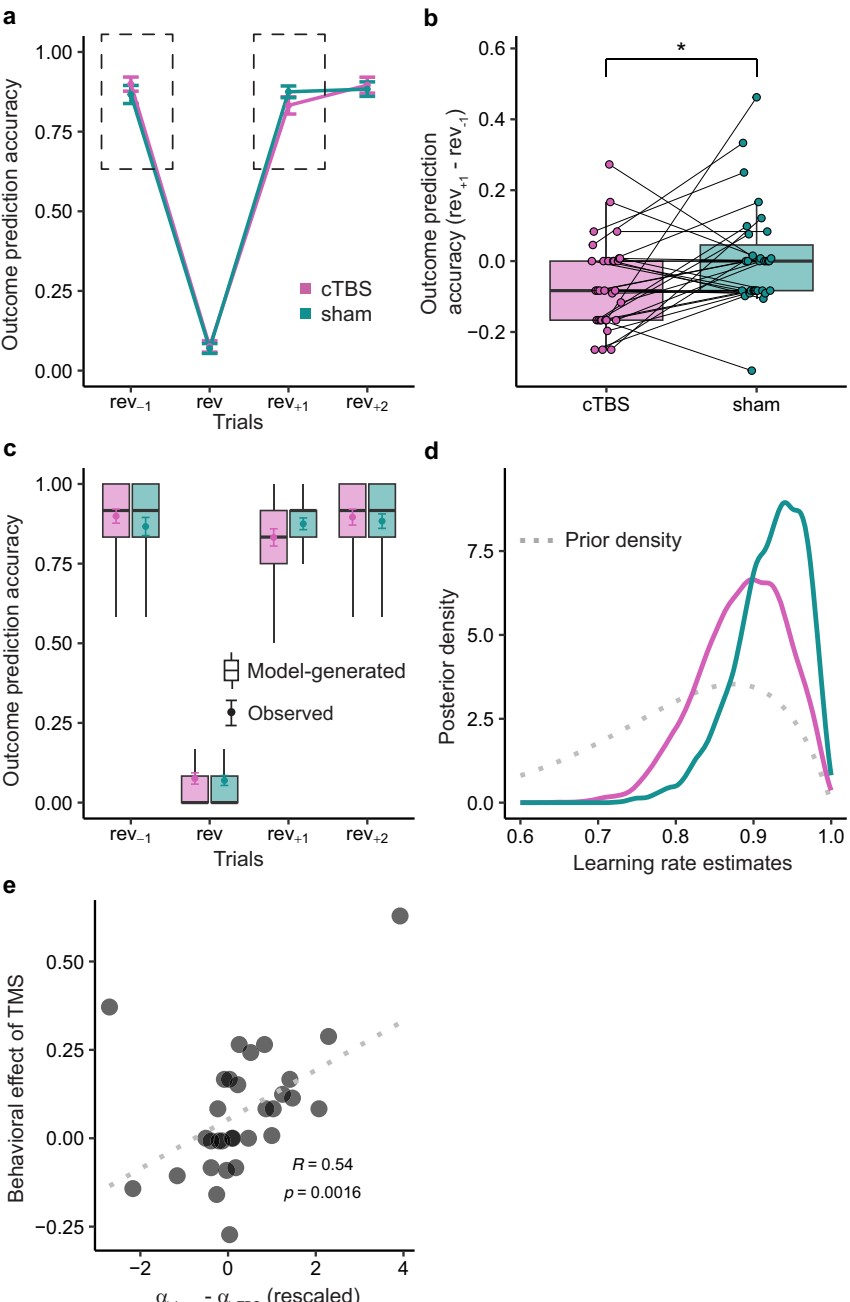

**Fig. 4 | cTBS disrupts outcome prediction accuracy. a** Behavioral outcome prediction accuracy for $rev_{-1}$, $rev$, $rev_{+1}$, and $rev_{+2}$ trials in the cTBS and sham sessions ($n = 31$). Data are presented as mean values ± SEM. **b** Comparison of the change in accuracy from $rev_{-1}$ to $rev_{+1}$ trials (i.e., dashed boxes in **a**), between cTBS and sham sessions. Each pair of dots connected by a line represents one subject ($n = 31$). Paired $t$ test, t(30) = 2.344, $p = 0.013$, one-sided. * denotes $p < 0.05$. **c** Model-generated outcome prediction accuracy (in boxplot) is overlayed with the observed accuracy (in error bars, data are presented as mean values ± SEM) at $rev_{-1}$, $rev$, $rev_{+1}$, and $rev_{+2}$ trials in the cTBS and sham sessions ($n = 31$). We used the identity learning model with session-wise learning rates to generate the accuracy predictions. **d** Posterior density of hyper learning rate estimates in cTBS and sham

sessions, obtained using a hierarchical Bayesian modeling method. The dotted line indicates the prior density used for estimating the hyper learning rate parameter. **e** Scatter plot showing the behavioral effect of TMS (difference in accuracy change ($re_{+1} - rev_{-1}$) between sham and cTBS), against the estimated learning rate difference, across 31 subjects. For illustration, the learning rate was logit transformed before calculating the difference. The gray dotted line denotes the linear fit between the behavioral effect of TMS and the learning rate difference. Spearman's rank correlation, R(29) = 0.54, $p = 0.0016$. **b**, **c** Box plots show center line as median, box limits as the first and third quartiles, whiskers as minimum to maximum values that are within 1.5 inter-quartile range from box limits. Rev: reversal; cTBS: continuous theta-burst stimulation. Source data are provided as a Source Data file.

change in accuracy (paired $t$ test, t(30) = 0.455, $p = 0.65$, two-sided).

To rule out effects of cTBS on the ability to perceive and identify odors, we asked subjects to perform an odor identification test after each scanning run. Overall, there was no significant difference in odor identification performance between the cTBS and

sham sessions (paired $t$ test, t(30) = 1.13, $p = 0.27$). However, we did observe a decrease in odor identification performance during the first run in the cTBS session relative to the sham session (paired $t$ test, t(30) = 2.53, $p = 0.017$). Nonetheless, this reduction in identification performance was not correlated with the decline in task performance (r = −0.047, t(29) = −0.25, $p = 0.80$), suggesting that

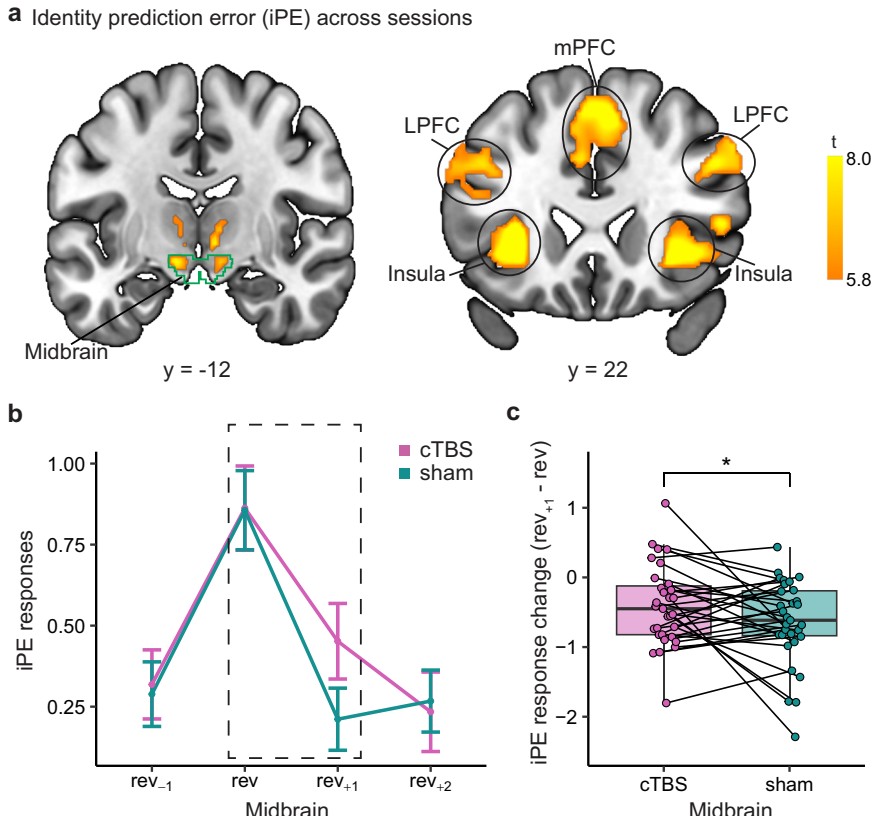

**a** Identity prediction error (iPE) across sessions

**Fig. 5 | cTBS alters neural responses to identity prediction errors in the midbrain. a** Brain areas showing stronger fMRI responses to reward outcomes on reversal trials compared to non-reversal trials. All clusters survive whole-brain FWE correction at $p_{FWE} < 0.05$. The green outline indicates the anatomical midbrain ROI. **b** Outcome-related fMRI responses in the midbrain ROI in cTBS and sham sessions across $rev_{-1}$, $rev$, $rev_{+1}$, and $rev_{+2}$ trials ($n = 31$, data are presented as mean values ± SEM). The dashed squares indicate the change of the response from $rev$ to $rev_{+1}$ shown in (**c**). **c** The change of outcome-related fMRI responses in the midbrain ROI from $rev$ to $rev_{+1}$ trials, in sham and cTBS sessions ($n = 31$). Each pair of dots connected by a line represents one subject. Paired $t$ test, t(30) = 1.846, $p = 0.037$, one-sided. * denotes $p < 0.05$. Box plots show center line as median, box limits as the first and third quartiles, whiskers as minimum to maximum values that are within 1.5 inter-quartile range from box limits. Rev reversal, iPE identity prediction error, mPFC medial prefrontal cortex, lPFC lateral prefrontal cortex, cTBS continuous theta-burst stimulation. Source data are provided as a Source Data file.

impaired odor perception cannot account for impairments in identity learning.

We next fitted our identity learning model to the behavioral data from the first run using a hierarchical Bayesian approach[34] (see "Methods"). We first compared a model with session-wise learning rates to a model with learning rates fixed across sessions. The model with session-wise learning rates (boxplot in Fig. 4c) accounted well for the observed outcome prediction accuracy (error bars in Fig. 4c) and provided a better account of behavioral accuracy than the model with fixed learning rates (deviance information criterion[35], DIC; session-wise learning rates 3585.9, and fixed learning rates 3620.2). Importantly, in line with our model simulations, the posterior density of learning rate estimates in the cTBS session was shifted towards lower values and had a larger range compared to the sham session (Fig. 4d). Additionally, the difference in learning rates between the sham and cTBS sessions was positively correlated with the behavioral effect of TMS, defined as the difference in changes in accuracy ($rev_{+1} - rev_{-1}$) between the two sessions (Spearman's rank correlation, R = 0.54, $p = 0.0016$, Fig. 4e). This is consistent with predictions of our identity learning model (Fig. 3b) and suggests that individual differences in the effect of TMS on changes in accuracy ($rev_{+1} - rev_{-1}$) can be explained by differences in learning rates.

Taken together, these results demonstrate that cTBS decreased outcome prediction accuracy after reversals ($rev_{+1}$) compared to pre-reversal trials ($rev_{-1}$) in the first run and thereby support the notion that

cTBS targeting the lateral OFC network impairs reward identity learning.

## cTBS alters neural responses to identity prediction errors in the midbrain

To investigate how disruption of lateral OFC activity affects neural signaling of iPEs, we first identified brain areas in which fMRI activity correlated with iPEs across both sessions. To increase statistical power, this and the following fMRI analyses used data from all three runs. In line with previous studies[10–12,36,37], we found significantly stronger fMRI responses to outcomes on reversal compared to non-reversal trials in the midbrain, as well as other cortical and subcortical regions including the medial frontal cortex, LPFC, and insula ($p_{FWE} < 0.05$, whole-brain FWE corrected, Fig. 5a, Supplementary Table 1). In addition, we found clusters in the lateral OFC that survived small volume correction (Supplementary Fig 5a).

Next, we evaluated the impact of cTBS on iPE signals by comparing fMRI responses in the midbrain between the cTBS and sham sessions (Fig. 5b). Following our model predictions (Fig. 3c), we estimated fMRI responses on $rev_{-1}$, $rev$, $rev_{+1}$, and $rev_{+2}$ trials and tested for differences on $rev_{+1}$ trials between cTBS and sham sessions. As predicted by the model with different learning rates, we found that fMRI responses to outcomes after the reversal were larger after cTBS (paired $t$ test, t(30) = 1.916, $p = 0.032$, one-sided), whereas they did not differ significantly on the reversal trial (paired $t$ test, t(30) = 0.049, $p = 0.519$,

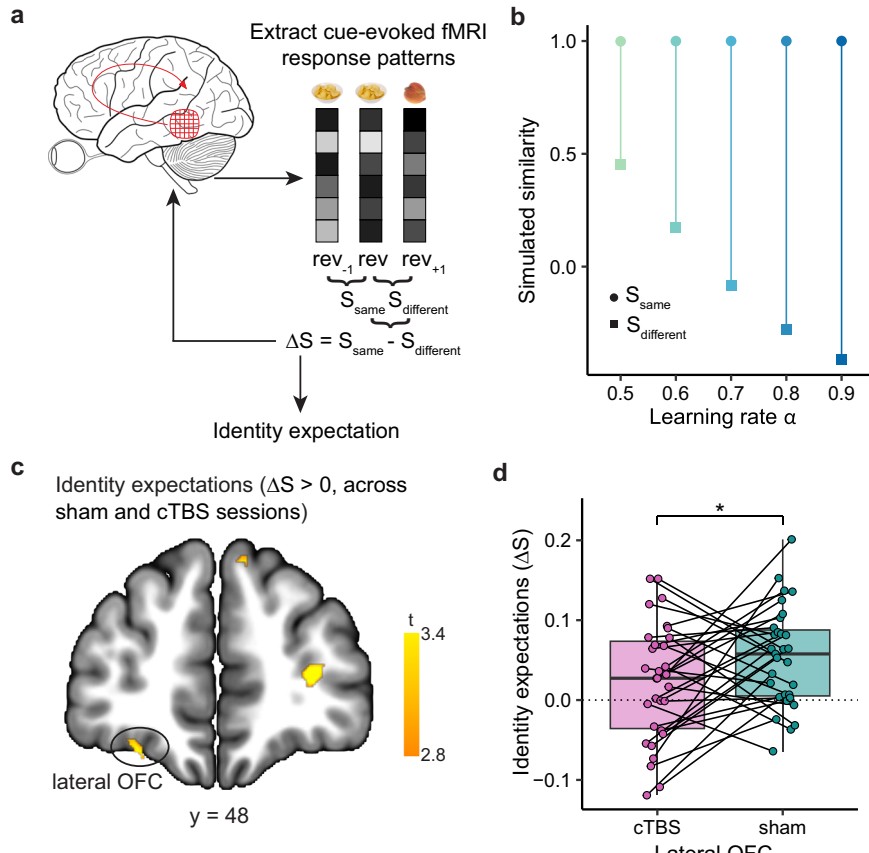

**Fig. 6 | cTBS disrupts reward identity expectations in the lateral OFC.**
**a** Illustration of the searchlight-based multi-voxel pattern similarity analysis. For each identity reversal, we calculated neural similarity (Pearson's correlation of voxel-wise beta values) between cue-evoked response patterns on *rev* and *rev*$_{-1}$ trials ($S_{same}$) and between *rev* and *rev*$_{+1}$ trials ($S_{different}$). The strength of identity expectation is defined as the difference $\Delta S$ between $S_{same}$ and $S_{different}$. **b** Simulated similarity of identity expectation strengths (V) between *rev*$_{-1}$ and *rev* trials ($S_{same}$, circles) and between *rev*$_{+1}$ and *rev* trials ($S_{different}$, squares), as well as their difference, under learning rates $\alpha$ from 0.5 to 0.9. **c** Lateral OFC cluster representing reward identity expectations across both sessions, using a lenient threshold of $p < 0.005$ (uncorrected). Peak coordinates: [−26, 48, −18], t(30) = 3.73; [−24, 30, −14], t(30) = 3.18. In a separate analysis using only data from the sham session,

representations of expected reward identity in the left lateral OFC ([−24, 44, −16], t(30) = 3.3, $p_{SVC\text{-}FWE}$ = 0.045; [−24, 36, −14], t(30) = 3.27, $p_{SVC\text{-}FEW}$ = 0.049) survived small volume family-wise error correction for multiple comparisons within the lateral OFC seed region. **d** Analysis of reward identity expectations in a lateral OFC ROI defined using clusters in (**b**) that overlapped with the targeted OFC seed region. Each pair of dots connected by a line represents one subject (*n* = 31). Wilcoxon signed rank test, V = 343, *p* = 0.032, one-sided. * denotes *p* < 0.05. Box plots show center line as median, box limits as the first and third quartiles, whiskers as minimum to maximum values that are within 1.5 inter-quartile range from box limits. Rev reversal, OFC orbitofrontal cortex, FWE-SVC family-wise error small-volume correction. Source data are provided as a Source Data file.

one-sided). Importantly, the difference between responses on *rev* and *rev*$_{+1}$ trials was significantly smaller in the cTBS compared to the sham session (paired *t* test, t(30) = 1.846, *p* = 0.037, one-sided, Fig. 5c), indicating that iPE responses in the cTBS session were slower to return to baseline after reversals. This result was further supported by linear mixed effect models that accounted for the effects of subjects and session orders, such that adding TMS (cTBS vs sham) as a predictor significantly improved the model fit (*p* = 0.013). In contrast, no significant differences between cTBS and sham sessions were found on the *rev*$_{-1}$ (paired *t* test, t(30) = 0.255, *p* = 0.40, one-sided) or *rev*$_{+2}$ trials (paired *t* test, t(30) = 0.236, *p* = 0.59, one-sided). Moreover, there was no significant correlation between the behavioral and neural effect of cTBS in the midbrain (r = 0.154, *p* = 0.41).

We conducted similar analyses in the cortical areas that were directly stimulated or indirectly targeted by our TMS protocol (LPFC and OFC, respectively). In the LPFC, decreases in fMRI responses from *rev* to *rev*$_{+1}$ trials were significantly smaller in cTBS compared to sham (paired *t* test, t(30) = 2.623, *p* = 0.0068, one-sided, Supplementary Fig 5c) and the fit of a linear mixed effects model was improved by including TMS as a factor (*p* = 4.6e−4). However, there was no significant difference on the *rev*$_{+1}$ trial (t(30) = 1.536, *p* = 0.067, one-sided)

or the *rev* trial (t(30) = 0.987, *p* = 0.17, one-sided). Interestingly, the effects of cTBS on changes in the midbrain and the LPFC were significantly correlated (r = 0.547, *p* = 0.0015, Supplementary Fig 5b), suggesting that cTBS affected iPE responses similarly in both regions. In the OFC, a similar effect was only significant in the left hemisphere (t(30) = 2.48, *p* = 0.0095, one-sided, Supplementary Fig 5d).

Overall, these findings show that cTBS modulated neural responses to identity prediction errors in the midbrain. Specifically, TMS targeting the lateral OFC caused fMRI responses to return to baseline more slowly after identity reversals. This is consistent with the idea that iPEs in the midbrain depend on activity in the lateral OFC network.

## cTBS disrupts reward identity expectations in the lateral OFC

A fundamental assumption of our experiment and model was that OFC-targeted cTBS would disrupt reward identity expectations in the lateral OFC. To directly test this possibility, we conducted a multi-voxel pattern similarity analysis on cue-evoked fMRI responses (Fig. 6a). We defined neural representations of identity expectations as the difference in correlation ($\Delta S$) between cue-evoked activity patterns from trials when the same ($S_{same}$) versus different ($S_{different}$) reward identify

was associated with a given cue. Specifically, for the cue-evoked activity pattern from each *rev* trial, we computed the correlation with the activity pattern from the preceding *rev*$_{-1}$ trial ($S_{same}$) and subtracted the correlation with the activity pattern from the following *rev*$_{+1}$ trial ($S_{different}$). This way, the trial prior to reversals serves as a baseline to measure identity expectations, analogous to using the pre-reversal trials as a baseline to measure behavioral adjustment from pre to post-reversals. Applying the same approach to identity expectations strengths from our model (Fig. 6b) illustrates that $\Delta S$ increases with higher learning rates, with $S_{same}$ consistently surpassing $S_{different}$, suggesting that cTBS should reduce $\Delta S$.

To define an unbiased ROI for comparing identity expectations between cTBS and sham, we performed a searchlight analysis to identify brain regions that represented reward identity expectations (positive values of $\Delta S$) using data from both the sham and cTBS session. Given our assumption that identity expectations would be disrupted by cTBS in half of the data, we used a lenient threshold ($p < 0.005$, uncorrected) for this analysis. Within the lateral OFC (Fig. 6c), we found significant clusters in both the left ([−26, 48, −18], t(30) = 3.73; [−24, 30, −14], t(30) = 3.18) and right hemisphere ([16, 40, −18], t(30) = 3.34; [14, 58, −16], t(30) = 3.41), but only the clusters in the left hemisphere overlapped with the OFC seed.

To examine whether cTBS disrupted reward identity expectations, we next analyzed activity patterns in lateral OFC clusters identified above that overlapped with the targeted OFC seed regions. Representations of reward identity expectations were significant in the sham (V = 442, $p = 4.49e{-}5$, one-sided) but not the cTBS session (V = 328, $p = 0.12$, one-sided). Importantly, neural representations of identity expectations differed significantly between the sham and cTBS session (V = 343, $p = 0.032$, one-sided, Fig. 6d). However, the cTBS effect on identity expectation in the lateral OFC was not significantly correlated with the behavioral effect of cTBS (r = −0.105, $p = 0.58$).

## Discussion

Work across species has shown that whereas the lateral OFC represents expectations about the identity of future rewards[2,3,19,21–23], activity in the dopaminergic midbrain responds to violations in such expectations[10,11,13,14]. Here we tested the causal contribution of the lateral OFC to neural signaling of iPEs, using a network-targeted cTBS approach to perturb activity in the lateral OFC network[26,32]. Verifying this targeted approach, cTBS decreased the global connectedness of the lateral OFC in the first run, indicating that TMS transiently altered neural processing in the lateral OFC. Paralleling these effects in the OFC, cTBS impaired behavioral outcome predictions in the first run of the trans-reinforcer reversal learning task, while altering iPE responses in the midbrain and disrupting neural representations of expected reward identity in the lateral OFC.

Associations between sensory cues and the identity of future rewards are key components of cognitive maps that can be used for model-based inference and learning[1,12,38]. By modulating activity in the OFC and measuring fMRI responses in the midbrain, our study reveals how the lateral OFC network and midbrain synergistically contribute to the learning and updating of such maps. Our findings suggest a neurobiological model of this learning process in which outcome-specific expectations in the lateral OFC[2–4,10,19] contribute to the computation of sensory prediction errors, which are signaled by the dopaminergic midbrain[3,39] and in turn update outcome-specific expectations in the lateral OFC.

This model is consistent with previous work suggesting that interactions between the OFC and midbrain support reward learning[40]. For instance, OFC lesions in rats alter value-related error signals in the ventral tegmental area (VTA)[39,41], and we have previously shown that iPE responses in the human midbrain correlate with changes in neural representations of expected reward identity in the lateral OFC[10]. The

current study extends these findings by demonstrating a causal link between lateral OFC representations and iPE signals in the midbrain.

Our results leave open the question whether iPEs are computed within the midbrain or in upstream regions. To compute prediction errors, an area needs information about what is expected and what is received. Dopamine neurons in the VTA have been proposed as a candidate for computing reward prediction errors[42], and the same inputs and mechanisms would put dopamine neurons in a good position to compute iPEs. For example, previous studies show that the lateral OFC provides excitatory inputs to GABA-ergic neurons in the VTA[43], which, in turn, inhibit VTA dopamine neurons[44]. Furthermore, dopamine neurons receive direct excitatory input about received rewards from sensory cortex and the lateral hypothalamus[45,46]. While these pathways are typically hypothesized to carry information about value, the upstream regions are known to also encode sensory features of outcomes[2,3,10,19–24], which could be used to signal iPEs. Our results showing that modulating activity in lateral OFC changes iPE responses in the midbrain are compatible with the idea that information about expected reward identity from the lateral OFC converges on dopamine neurons with information about received reward to compute identity prediction errors. However, we cannot rule out the possibility that other, additional regions are involved in the computation of iPEs.

In addition to the midbrain, we found identity prediction error responses in other areas, including the striatum, mPFC, thalamus, insula, and lateral OFC. Responses to identity errors in the lateral OFC are consistent with previous human fMRI studies[10,11,23] but contrast with what has been observed using electrophysiological recordings in rats[47]. This discrepancy could reflect differences between species or training requirements but seems most likely to be due to the different neural recoding methods. Specifically, single-unit recording approaches measure spiking activity and are biased to sample from large pyramidal neurons, whereas BOLD responses are more closely related to local field potentials (reflecting presynaptic activity) than multi-unit responses[48]. This suggests that fMRI responses to iPEs in the lateral OFC could reflect the influence of input from dopaminergic neurons on activity in neural populations that are poorly sampled by unit recording approaches (e.g., local interneurons) or on subthreshold events that are not visible in spiking activity.

In line with previous suggestions that OFC represents the identity of future rewards[3,19,21,33] and other task states[24,49–51], our multivoxel pattern similarity analysis revealed representations of expected reward identity in the lateral OFC. These representations were disrupted by cTBS, showing that our TMS approach not only modulated the connectivity in the targeted OFC area but also diminished the strength of neural representations in this region. We speculate that the disruption of these reward identity expectations caused the change in identity PEs observed in the midbrain.

We modeled the effects of disrupted OFC function on signaling reward identity expectations as a reduction in learning rates. Both the behavioral and neural effects of cTBS were highly consistent with this model. Importantly, however, we do not suggest that the function of the OFC is to regulate learning rates, but that disrupting OFC function has system-level effects that can be conveniently modeled by reducing learning rates.

The network-targeted cTBS protocol used here differed from our previous studies[26,32] in two ways. First, we used a placebo coil for the sham session instead of lowering stimulation intensity. Second, and more importantly, we stimulated both hemispheres instead of only the right side. The current findings parallel our earlier observations[26], showing that cTBS decreases global connectedness in the lateral OFC, albeit on both sides, as intended. In addition, the current results also provide important information on the duration of these effects, revealing that this protocol disrupts OFC connectivity for approximately 23 min (elapsed time from the end of TMS to the end of the first run: 22.97 ± 1.97 min). However, in contrast to our previous study[26], we

also observed an effect on the directly stimulated area in LPFC. This highlights the need to further investigate the mechanisms of network-targeted TMS, including its effects on both the indirectly targeted and directly stimulated areas[52].

Taken together, our findings support the idea that representations of expected outcome identity in the lateral OFC contribute to iPEs responses in the midbrain by providing the identity predictions necessary for computing the error signal. Thus, our results support a systems-level model of identity learning in which midbrain iPEs are generated by comparing incoming sensory information with outcome identity expectations stored in the OFC.

## Methods
### Subjects
Forty-two healthy right-handed human subjects with no history of psychiatric or neurological disease gave informed written consent to participate in this study. Thirty-five completed all sessions. Of these, data from four subjects were excluded from analysis: two because they did not tolerate cTBS at 80% rMT such that intensity was adjusted to 50% rMT, and two due to poor odor identification performance (less than 80% correct on average). All results reported in this manuscript are based on the remaining 31 subjects (11 male, ages 19–42, mean = 26.39, SD = 6.04). The experimental protocol was approved by the Northwestern University Institutional Review Board and registered (NCT04926961).

### Study design
The study consisted of four separate sessions (Fig. 1d), described in detail below. Day 2 was scheduled on average 4.52 d (SD = 4.39 d) after Day 1 (Day 1 and Day 2 sessions occurred on the same day for six subjects for ease of scheduling). The average time between Day 2 and Day 3 was 9.23 d (SD = 8.24 d). Day 3 and Day 3 were the main study days and were scheduled, on average, 8.29 d (SD = 2.71 d) days apart. On these days, subjects received either sham or active cTBS, with the order of sham and cTBS counterbalanced across subjects and sex. 17 subjects (5 males, 12 females) out of 31 received sham on Day 3. For all sessions, subjects were instructed to arrive in a hungry state, having fasted for at least four hours prior to testing. Subjects were compensated with $20 per hour of behavioral testing, $30 per hour of TMS procedure, and $40 per hour of fMRI scanning.

### Day 1: screening session
After informed consent and screening for eligibility, subjects rated the pleasantness of ten food odors (Fig. 1d). Food odors were provided by International Flavors and Fragrances (New York, NY), and included four savory food odors (pot roast, potato chip, sauteed onion, and barbecue), three sweet food odors (caramel, chocolate, and gingerbread), and three fruity food odors (strawberry, peach, and pineapple).

On each trial, subjects smelled one food odor for two seconds and then rated how much they liked the odor on a scale from "Most Disliked Sensation Imaginable" to "Most Liked Sensation Imaginable." All ratings were made on visual analog scales through a scroll wheel and keyboard button press. Each food odor was presented three times in a pseudo-randomized order and ratings were averaged per odor. Based on these ratings, we selected three odors (one from each of the three categories) that were rated as pleasant (above neutral) and most closely matched. These three odors were used as rewards for the trans-reinforcer reversal learning task on Day 3 and Day 4. Subjects were excluded from further participation if no such three odors were found. Subjects then rated the intensity and pleasantness of the three selected odors. The scale of the intensity rating was from "Undetectable" to "Strongest Imaginable." Subjects were also tested on their ability to discriminate between each pair of odors and rated the similarity between two odors to make sure that they were able to distinguish between the odors. Supplementary Fig 1 displays data on how subjects rated the three selected odors in terms of their pleasantness, intensity, and similarity, along with their ability to discriminate between different odors.

We used a custom-built computer-controlled olfactometer to deliver food odors with precise timing directly to nasal masks worn by subjects. This olfactometer directed medical-grade air through the headspace of amber bottles containing liquid solutions of food odors, maintaining a constant flow rate of 3.2 L/min. Using two independent mass flow controllers (Alicat, Tucson, AZ), the olfactometer enabled the dilution of odorized air with odorless air. Throughout the experiment, a constant stream of odorless air was delivered to subjects' noses. Odorized air was mixed into this airstream at specific time points, while ensuring no impact on the overall flow rate and preventing any alteration in somatosensory stimulation.

### Day 2: MRI and TMS resting motor threshold session
We acquired a T1-weighted structural MRI scan to assist TMS neuro-navigation and an 8.5 min resting-state fMRI scan (250 volumes, TR = 2 s) to individually define OFC-targeted cTBS coordinates (see below). We then measured rMT by administering single TMS pulses to the hand area of the left motor cortex. rMT was defined as the lowest percentage of stimulator output required to evoke 5 visible thumb movements from 10 pulses. For 3 subjects, rMT was determined using TMS pulses over the right motor cortex, due to difficulties evoking isolated finger movements on the left.

### Days 3 and 4: OFC-targeted cTBS sessions
Our TMS protocol was modified from our previously established OFC network-targeted protocol[26,32]. The key difference is that stimulation was applied to both hemispheres instead of just the right. TMS was applied at an intensity of 80% rMT using a cTBS protocol consisting of three-pulse 50 Hz bursts delivered every 200 ms (5 Hz) for a total of 600 pulses (~40 s)[53]. TMS pulses were delivered using a MagPro X100 stimulator with a MagPro Cool-B65 A/P butterfly coil (MagVenture). Previous work has demonstrated that this cTBS protocol at 80% MT has inhibitory aftereffects which persist for 50–60 min over primary motor cortex. We administered 600 pulses to both left and right hemisphere[54], alternating the order across subjects, with 16/31 starting on the left. The order was consistent per subject for both sessions. Immediately after the final pulse of cTBS on the second side, the time was recorded, and subjects moved across the hall into the MRI to perform three runs of the trans-reinforcer reversal learning task during fMRI data acquisition (see below). The first run of the task started 8.77 min (SD = 1.97 min) after the end of TMS. No significant difference in this duration was observed between cTBS and sham sessions (t(30) = 1.63, p = 0.11, paired t test).

Similar to our previous work[26,32], the target coordinates were defined as the locations in the left and right LPFC with the strongest functional connectivity with the left and right OFC seed regions (see details below). The figure-of-eight coil was tilted so that its long axis was approximately perpendicular to the long axis of the middle frontal gyrus. Whereas cTBS was delivered by positioning the active side of the A/P coil to modulate neural tissue, sham cTBS was applied with the placebo side of the A/P coil, producing similar somatosensory and auditory experiences for the subject without modulating neural tissue. Electrodes were placed on subjects' forehead and direct current stimulation was applied in synchrony with the TMS pulses to mask TMS effects and enhance the similarity between cTBS and sham sessions.

Subjects were informed about potential muscle twitches in the face, eyes, and jaw during the simulation. To test for tolerability, two single pulses were applied over the stimulation coordinates before administering cTBS. Two subjects reported discomfort with the intensity of 80% rMT, so the intensity was reduced to 50% rMT and their data were excluded from analysis.

We assessed subjects' discomfort and perceived stimulation intensity after each of the two TMS sessions. The cTBS session was generally perceived as more uncomfortable and intense compared to the sham session. On a scale ranging from 0 (not uncomfortable at all) to 10 (extremely uncomfortable), mean discomfort ratings were 2.69 and 5.65 for the sham and cTBS session, respectively (t(30) = −7.92, $p$ = 7.72e−09). Similarly, on a scale ranging from 0 (not strong at all) to 10 (extremely strong), mean intensity ratings were 3.44 and 6.71 for the sham and cTBS session, respectively (t(30) = −9.29, $p$ = 2.45e−10). During debriefing at the end of the study, 25 out of the 31 subjects reported having noticed differences between the two TMS sessions. Also, when asked, 18 out of 29 subjects were able to correctly name on which session they received sham or cTBS. However, 24 subjects reported that they had not considered the real or sham nature of our TMS procedure before the debriefing interview. We did not collect these data from two subjects. Overall, it appears that despite our attempts to obscure the somatosensory differences between the sham and cTBS sessions, differences between cTBS and sham TMS could not be fully concealed. Nevertheless, it is worth noting that most subjects did not consider this question during the experiment. For each of the analyses presented in the manuscript, we considered discomfort and perceived TMS intensity as potential confounding variables and performed control analyses to test whether they could account for the results (see Supplementary Information). All control analyses show that the TMS effects reported here cannot be explained by subjects' physical discomfort or perceived TMS intensity.

## Coordinate selection for lateral OFC network-targeted TMS

The stimulation coordinates on the bilateral LPFC were individually determined based on the resting-state fMRI connectivity[52,55,56] from Day 2, using central-lateral OFC seed regions in each hemisphere. The central-lateral OFC was selected based on previous research linking activity in this region to outcome-specific expectations[19,22,33] and strong functional connectivity of this region with the LPFC in a large resting-state dataset ($N$ = 1000, neurosynth.org). In other words, we focused on an area of lateral OFC that is relevant to our hypotheses regarding the cognitive function and that is a "good" indirect target in the sense of having robust connectivity to stimulation-accessible locations within LPFC. Specifically, we used coordinates in the central-lateral OFC (right: x = 28, y = 38, z = −16; left: x = −28, y = 38, z = −16), and in the LPFC (right: x = 48, y = 38, z = 20; left: x = −48, y = 38, z = 20). We first generated spherical masks of 8-mm radius around these four coordinates in MNI space, each inclusively masked by the gray matter tissue probability map provided by SPM12 (thresholded at >0.1). We then transformed these four masks to each subject's native space using the inverse deformation field generated during the normalization of the T1 anatomical image (see below). Resting-state fMRI data were realigned, resliced, and smoothed with a 6 mm Gaussian kernel. We then specified two resting-state fMRI functional connectivity analyses (one per hemisphere) for each subject, using individual OFC masks as the seed region and motion parameters from the realignment as regressors of no interest. Finally, stimulation coordinates were defined as the voxels within the left and right LPFC masks with the strongest functional connectivity to the left and right OFC seed regions, respectively. We used infrared MRI-guided stereotactic neuronavigation (LOCALITE) to apply stimulation to these two LPFC coordinates.

## Trans-reinforcer reversal learning task

The trans-reinforcer reversal learning task used on Day 3/4 (Fig. 1a) required subjects to learn the association between visual cues and the food odor rewards that were individually chosen based on pleasantness ratings from Day 1. Two abstract visual symbols were randomly chosen for each subject and used throughout the experiment (only one is shown in Fig. 1a). Each trial started with one of the two visual cues displayed on the screen for 1 s, followed by the presentation of

three abbreviated odor names. The subject was asked to indicate which odor they were expecting to receive, by pressing one of three buttons corresponding to the position of the three odor names on the screen (the order of the odor names on the screen was randomized across trials). If the subject responded within 3 s, the odor names turned gray, the cue disappeared with a fixation cross remaining on the screen for the following 4 s. Then, the odor associated with the current cue was delivered, signaled by the central fixation cross turning blue for 2 s cuing the subject to sniff. If no response was made within 3 s, "TOO SLOW" appeared on the screen for the following 4 s, followed by the delivery of the odor. The odor delivery was followed by a 2–4 s inter-trial interval pseudo-randomly sampled from a uniform distribution. The task was presented and behavioral data were acquired using the Cogent 2000 toolbox (v1.32) in Matlab (R2016b).

In each run, the association between cues and odors reversed six times per cue, independently for each cue, totaling 12 reversals (Fig. 1b). The number of trials between each reversal varied from three to five, preventing subjects from knowing when the next reversal would occur. This also allowed for each trial to be classified as reversal trial ($rev$), one or two trials after a reversal ($rev_{+1}$, $rev_{+2}$), or one trial before a reversal ($rev_{-1}$). The trials from the two cues were randomly interleaved, resulting in 64 trials per run and increasing the difficulty of predicting when the next reversal would occur.

After each run of the trans-reinforcer reversal learning task, subjects completed an odor identification task to control for possible effects of TMS on olfactory identification performance. On each trial, one of the three odors was delivered, and subjects had to select the correct odor name via a corresponding button press. Each odor was presented twice and performance was averaged across all trials. Data from two subjects was excluded from further analysis due to poor overall performance on the identification task (less than 80% correct). Paired $t$ tests were conducted to compare subjects' odor identification performance after cTBS and sham TMS.

## Behavioral data analysis

In analyzing behavioral data from the trans-reinforcer reversal learning task, we removed all trials in which subjects did not respond within 3 s and trials with response times outside the range of ±3 standard deviation from the mean response time for each subject. This resulted in the exclusion of 2.48% of trials from the behavioral analysis. We analyzed the accuracy of subjects' outcome predictions before, during, and after each reversal trial. In addition to comparing performance on cTBS and sham sessions across all three runs, we specifically focused on the first run to capture any transient effects on behavioral performance. The analysis and visualization of behavioral data (Fig. 4, along with ROI-based fMRI data in Fig. 2c, Fig. 5b, c, Fig. 6d) were conducted in R (4.1.2)/ RStudio (2022.07.1) with packages dplyr (1.0.8), plyr (1.8.6), and ggplot2 (3.4.1).

## Identity learning model

This identity-based learning model[10,11] is grounded in the delta learning rule and represents reward identity received on trial $t$ as a vector $\vec{I}_t$. The model updates the identity expectation strength $\vec{V}_t$ based on the error vector $\vec{\delta}_t$, which reflects the prediction error for each reward identity, such that

$$\vec{\delta}_t = \vec{I}_t - \vec{V}_t, \tag{1}$$

$$\overrightarrow{V_{t+1}} = \overrightarrow{V_t} + \alpha \overrightarrow{\delta_t} \tag{2}$$

The learning rate $\alpha$ affects the rate of updating the expectation strength vectors across trials, whose effect can be observed in Fig. 3a. In the simulation, the accuracy of outcome predictions (Fig. 3d) was calculated by transforming the identity expectation strengths into

probabilities using a softmax function

$$p_{it} = e^{\theta V_{it}} / \sum_z e^{\theta V_{zt}} \qquad (3)$$

Where $\theta$ is the inverse temperature parameter that controls the degree of randomness in the choice. We used a fixed $\theta = 5$ in the model simulation in Fig. 3b. $V_{it}$ denotes the expectation strength on the odor identity $i$ at trial $t$. A scalar term $e_t$ of identity PEs was defined as the sum of the absolute values of the elements of the error vector $\vec{\delta}_t$, such that

$$e_t = \sum_i |\delta_{it}|. \qquad (4)$$

Hence $e_t$ quantifies the overall discrepancy between the actual outcome and the representation weights across three reward identities and serves as an important bridge between the identity learning model and the fMRI responses to iPEs. In Fig. 3a, for illustrative purposes, the magnitude of $e_t$ was normalized within each condition by

$$f(x) = \frac{x - \min(x)}{\max(x) - \min(x)}, \qquad (5)$$

such that the lowest and highest $e_t$ values are set to 0 and 1, respectively.

## Fitting learning models to behavioral accuracy

We fitted the identity learning model to the behavioral data across all trials from the first run, using hierarchical Bayesian parameter estimation with a hierarchical structure on the learning rate parameter, such that

$$\alpha_{j,c} \sim Beta\left(\alpha_{\mu_c}\kappa, \left(1 - \alpha_{\mu_c}\right)\kappa\right), \qquad (6)$$

where $\alpha_{j,c}$ denotes the learning rate for session c (c = cTBS or sham) and subject $j$. The beta function $Beta(\mu\nu, (1 - \mu)\nu)$ is parameterized using the "mean" ($\mu$) and "sample size" ($\nu$) parameters, so that $\alpha_{\mu_c}$ denotes the hyper learning rate in session c. The prior distributions for $\alpha_{\mu_c}$ and $\kappa$ are specified by

$$\alpha_{\mu_c} \sim Beta(8, 2), \qquad (7)$$

$$\kappa \sim Gamma(1, 0.1), \qquad (8)$$

where $Gamma(r, \lambda)$ denotes a gamma distribution with the shape parameter $r$ and rate parameter $\lambda$, and $Beta(a, b)$ denotes a beta distribution with the two shape parameters $a$ and $b$. We used an informative prior distribution of $Beta(8, 2)$ to express our prior belief that the learning rate distribution should be heavily skewed towards high values near one, given that our task was simple. In addition, we specified a prior distribution of Gamma (5, 1) for the inverse temperature parameter $\theta$. To help assess the effect of cTBS on the learning rate, we created a baseline model with the identical model structure and priors except for the absence of session-wise learning rate parameters $\alpha_{\mu_c}$ and $\alpha_{j,c}$.

We estimated the hierarchical Bayesian model in JAGS[57] using the R2jags package[58] in R. We sampled from the posterior distributions for 5000 iterations with 2000 burn-in and three chains, resulting in 9000 samples per model. We ensured convergence by examining the $\hat{R}$ under 1.1[59] for each model and each parameter. The maximum a posterior (MAP) estimates of the posterior samples were calculated for all subject-level parameters and hyperparameters. We compared hierarchical models using the deviance information criterion (DIC) from the R2jags output, with lower DIC values indicating better model

performance[35]. To get an absolute sense of how well the identity learning model captured subjects' outcome prediction accuracy across trials, we performed a posterior predictive check by simulating the learning models 1000 times using the MAP estimates for each subject and compared the model-simulated prediction accuracy with observed data. We also correlated the difference of learning rates between cTBS and sham sessions with the accuracy difference on trials $rev_{+1}$ across subjects.

## Sniff recording and analysis

We measured subjects' nasal airflow (i.e., sniffing) directly at the nose through a nasal mask, using a respiratory flow head and a spirometer. We recorded nasal airflow data using PowerLab equipment (ADInstruments, Dunedin, New Zealand) at a sampling rate of 1 kHz. Airflow traces for each fMRI run were preprocessed, including smoothing with a 250 ms moving window, down-sampling to 10 Hz, high-pass filtering (50 s cutoff) to eliminate slow signal drifts, normalization by subtracting the mean and dividing by the standard deviation across the run trace, and a final down-sampling to 0.5 Hz to align with the TR. We computed sniff volume as the integral of the nasal airflow. Nasal airflow, sniff volume, and their squared values were included as nuisance regressors in all fMRI analyses.

## MRI data acquisition

MRI data were acquired on a Siemens 3 T PRISMA system equipped with a 64-channel head-neck coil. For resting-state fMRI on Day 2, 250 echo-planar imaging (EPI) volumes were acquired with a parallel imaging sequence with the following parameters: repetition time, 2 s; echo time, 22 ms; flip angle, 80°; multi-band acceleration factor, 2; slice thickness, 2 mm, no gap; number of slices, 58; interleaved slice acquisition order; matrix size, 104 × 96 voxels; field of view, 208 mm × 192 mm. The functional scanning window was tilted ~30° from axial to minimize susceptibility artifacts in the OFC[60]. In addition, a 1 mm isotropic T1-weighted structural scan was acquired for neuronavigation during TMS and to aid spatial normalization.

On Day 3 and Day 4, subjects performed three runs of the transreinforcer reversal learning task while fMRI data were acquired with the same parameters as the resting-state scan. Each run lasted 14.4 min and consisted of 430 EPI volumes, covering all but the dorsal portion of the parietal lobes. To aid coregistration between the functional scans and the anatomical image, ten whole-brain EPI volumes were acquired for each subject on Day 2, Day 3, and Day 4, using the same scanning parameters as described above except covering the whole brain (95 slices) with a repetition time of 3.15 s.

## Imaging data preprocessing

We preprocessed the imaging data using the Statistical Parametric Mapping (SPM12) software (www.fil.ion.ucl.ac.uk/spm/) in Matlab (R2020b, Mathworks Inc). For each subject, we corrected for head movement by realigning all functional EPIs from all runs and sessions to the first acquired image. We also realigned and averaged the ten whole-brain EPIs for each subject, and coregistered the mean whole-brain EPI to the T1 anatomical image. We then coregisterd the mean functional EPI to the mean whole-brain EPI and applied the transformation parameters to all functional EPIs. For spatial normalization, we normalized the T1 anatomical image from each subject to the Montreal Neurological Institute (MNI) space using the six-tissue probability map provided by SPM12, and then applied the resulting deformation fields to the functional EPIs to transform them into MNI space. Finally, we smoothed the normalized functional EPIs with a 6 mm full-width half maximum Gaussian kernel in all three spatial dimensions. For multivariate imaging data analysis, we instead applied a Gaussian kernel with 2 mm size to the resliced, coregistered, and normalized images for maintaining sufficient spatial information.

## Global connectedness analysis

To validate the network-targeted TMS protocol, we computed voxel-wise maps of global connectedness[26]. This measure assesses the average connectivity between the fMRI activity in a given voxel and all other gray matter voxels. Unlike[26], who used resting-state fMRI, we utilized the task-based fMRI data, because we did not collect resting-state fMRI scans after TMS. To increase computational efficiency, we resampled functional EPIs to 3 mm isotropic resolution and restricted the analysis to gray matter voxels (tissue probability map, thresholded at > 0.1).

Prior to computing global connectedness, we regressed out nuisance regressors from the activity time course of each gray matter voxel. These regressors included: the mean global signal in all gray matter voxels, all white matter voxels, all cerebrospinal fluid (CSF) voxels, linear drift, and a constant baseline, as well as the six volume-wise realignment parameters (three translations, three rotations) estimated during motion correction; the derivative, square, and the square of the derivative of each realignment regressor; the absolute signal difference between even and odd slices, and the signal variance across slices in each functional volume (to account for fMRI signal fluctuation due to within-scan head motion); the squares, derivatives, and squared derivatives of these two within-volume measures; the four sniff trace regressors (see **sniff recording and analysis**); and additional dummy regressors as needed to account for individual volumes with particularly strong head motion. White matter and CSF masks were defined using their respective tissue probability maps (thresholded at >0.9).

We regressed out all above regressors through a linear filtering process. We denote the number of volumes involved in the analysis as nvol, the number of gray matter voxels in 3 mm space as nvox, and the number of above regressors as nreg. We created a filter matrix F containing the above regressors of dimension (nvol*nreg), where all regressors except for the constant were z-scored. We also organized fMRI responses of each gray matter voxel in a matrix B of dimension (nvol*nvox). Then the weight W can be obtained from the least square solution of a linear regression that predicts B using F as

$$W = (F'F)^{-1}(F'B) \qquad (9)$$

where (.)' denotes transpose of a matrix and (.)$^{-1}$ denotes the inverse of a matrix. Global connectedness was calculated based on the residual matrix R after removing the variance explained by the filter matrix F from the fMRI responses, where

$$R = B - FW \qquad (10)$$

For each subject and each run, we calculated the Pearson correlation between each gray matter voxel and every other gray matter voxel, and then averaged the absolute Fisher's Z-transformed correlations for each voxel to obtain the voxel-wise global connectedness measure.

We obtained a whole-brain map of the effect of cTBS on global connectedness by comparing the global connectedness values from the cTBS and sham session, per run. The resulting difference maps were smoothed with a 6 mm FWHM kernel. To identify early effects of OFC network-targeted cTBS comparable to our previous study using resting-state fMRI[26], we focused on data from the first run. Voxels in the OFC and LPFC with reduced global connectedness after cTBS compared to sham were identified using one-sample $t$ tests, FWE-SVC corrected for the OFC and LPFC ROIs that were used in the connectivity analysis to determine TMS coordinates. Visualization of group-level effects of TMS on global connectedness (Fig. 2a, b), iPE (Fig. 5a), and identity expectation (Fig. 6c) are conducted using MRIcroGL(version 12.6).

To investigate how the effects of cTBS on global connectedness change over time, we conducted ROI analyses in the OFC and LPFC (same ones as used for FWE-SVC above), averaging across both hemispheres. We divided each run into two parts and calculated the global connectedness for each of the six time bins. We performed Wilcoxon signed-rank tests for each ROI in each time bin to determine whether cTBS decreased global connectedness relative to sham[61]. Additionally, we ran linear mixed effect regression models to analyze if the global connectedness after sham or cTBS changed over time. We assessed the difference in the linear trend of time between cTBS and sham by testing whether including interaction effects of time and session beyond the main effects improved the model fits. An improved model fit would suggest a significant interaction effect and, therefore, a difference in the linear trend between cTBS and sham.

Because the OFC and LPFC are strongly functionally connected, it is possible that changes in OFC connectedness were entirely driven by effects of TMS on the LPFC rather than the OFC. To rule out this possibility, we conducted an additional analysis in which we eliminated any contribution from the LPFC on global connectedness estimates in the OFC by excluding voxels within the LPFC ROI from the whole-brain gray matter voxels. This analysis yielded nearly identical results to the original analysis. Thus, the global connectedness effects observed in the OFC were not substantially driven by effects of TMS on the LPFC, suggesting that TMS altered the connectivity of the OFC beyond its connection to the LPFC.

## Univariate test for iPE fMRI signals

To examine neural responses to iPEs and how they were modulated by cTBS, we constructed subject-level event-related GLMs using regressors convolved with a canonical hemodynamic response function (HRF), and time-locked to the onset of the cue presentation and odor delivery. We modeled four different odor delivery conditions: reversal trials ($rev$), one trial before ($rev_{-1}$), one trial after ($rev_{+1}$), and two trials after a reversal ($rev_{+2}$). Only trials with valid responses were included: for non-reversal trials, correct outcome predictions were considered valid; for reversal trials, trials were considered valid if subjects predicted the odors received on trial $rev_{-1}$.

To increase the statistical efficacy, the odor delivery regressors from all three runs from a given session were concatenated. The cue-locked regressors from all six runs were combined into a single regressor. As a result, each subject-level GLM contained nine task-related regressors (sham $rev_{-1}$, sham $rev$, sham $rev_{+1}$, sham $rev_{+2}$, cTBS $rev_{-1}$, cTBS $rev$, cTBS $rev_{+1}$, cTBS $rev_{+2}$, all cues), along with nuisance regressors.

Nuisance regressors were the same as those used in the **Global connectedness analysis**. These nuisance regressors were concatenated across six runs, and six run-wise constant terms were added to account for baseline differences between runs. We computed single-subject contrast images, comparing fMRI responses associated with the onset of odor delivery on reversal trials with responses on non-reversal trials across both sessions ($rev$ vs. [$rev_{-1}$, $rev_{+1}$, $rev_{+2}$]).

For voxel-wise group analysis, we used an explicit mask consisting of voxels in the gray matter and the midbrain. Gray matter voxels were identified using the tissue probability map provided by SPM12 (thresholded at >0.1, excluding the cerebellum), and midbrain voxels were identified using a probabilistic atlas (thresholded at > 0.5)[62]. Significance threshold for voxel-wise $t$ tests was set at $p < 0.05$, family-wise error (FWE) corrected for multiple comparisons across the whole brain at the voxel level. For the OFC, FWE small volume correction (FWE-SVC) was performed in the OFC seed ROIs used to determine OFC network-targeted TMS coordinates (Fig. 1c).

We created three functional ROIs (midbrain, LPFC, and OFC) based on the contrast of reversal > non-reversal from both cTBS and sham sessions. For the LPFC and midbrain (Supplementary Fig 5a), we identified clusters of voxels around the peak activations (left midbrain:

[−10, −24, −10]; right midbrain: [10, −14, −10]; left LPFC: [−40, 6, 40]; right LPFC: [46, 22, 30]) that survived whole-brain correction at $p_{FWE} = 0.05$ with a voxel-extend threshold of 25. The functional midbrain ROI was further masked with the anatomical midbrain mask[62] thresholded at 0.5 to exclude adjacent voxels not part of the midbrain. For the OFC, the functional ROI (Supplementary Fig 5a) included voxels at $p < 0.001$ (uncorrected) from clusters around peak voxels that survived $p_{FWE-SVC} = 0.05$ for the OFC (left OFC: −22, 30, −14]; right OFC: [20, 42, −16]). One-sided paired $t$ tests were conducted to examine the difference in fMRI responses between cTBS and sham sessions.

To account for the order of the two experimental sessions (counter-balanced across subjects), we confirmed results in Fig. 5c using linear mixed effect models. For this, we compared the model fits of two linear mixed effect models: a full model that included effects from subjects, session orders, and TMS condition (cTBS vs sham), and a reduced model without the effect of TMS condition. We also conducted independent $t$ tests (two-sided) exploring if the cTBS effect on iPE signals was related to any across-subject factors such as subjects' sex. We found that the average effect of cTBS on iPE responses was smaller in females compared to males in the LPFC (t(14.33) = 2.86, $p = 0.012$, Supplementary Fig 5b), but not the midbrain (t(13.26) = 1.86, $p = 0.086$) or the left OFC (t(17.17) = 1.29, $p = 0.21$). Other across-subject factors we tested (using two-sided independent $t$ tests or Pearson's correlation tests) include TMS session order (cTBS or sham first), TMS intensity, and age, but we found no significant effect on those factors (all $p > 0.4$).

### Multivoxel pattern similarity analysis

To test for representations of expected reward identity, we conducted a multivoxel pattern similarity analysis. In a first step, we estimated single-trial beta values time-locked to the cue onsets using the "least squares separate" approach[63] that was designed to avoid the multicollinearity problem. For each trial, we specified a GLM with the regressor of interest being the "cue" onset of the current trial, and other regressors including the cue onsets of all other trials, "odor outcome" onsets of the current trial, "odor outcome" onsets of all other trials, and nuisance regressors (as above). We estimated the GLMs on resliced, coregistered, and normalized functional EPIs, smoothed with a 2 mm FWHM Gaussian kernel.

We then computed the difference in correlations ($\Delta S$) between activity patterns from the *rev* and *rev*$_{-1}$ trials ($S_{same}$) and between activity patterns from the *rev* and *rev*$_{+1}$ trials ($S_{different}$) as a neural measure of identity expectations. Specifically, for each identity reversal, we computed the Pearson's correlation between the patterns of cue-evoked single-trial beta estimates from the *rev* and *rev*$_{-1}$ trials, and between the cue-evoked patterns from the *rev* and *rev*$_{+1}$ trials. The correlation coefficients were Fisher's z-transformed, subtracted (corr(*rev*, *rev*$_{-1}$) minus corr(*rev*, *rev*$_{+1}$)), and averaged across reversals per session to obtain reward identity expectations. In this analysis, we included all trials from the experiment irrespective of subjects' behavioral response. The total number of reversals per session was 36 (i.e., 6 reversals per cue x 2 cues x 3 runs).

We used a whole-brain searchlight approach[64] with a 2 voxel radius (i.e., 4 mm) and mapped average $\Delta S$ onto the center voxel of each searchlight, resulting in whole-brain map of $\Delta S$ for each session and each subject. The resulting $\Delta S$ maps were smoothed with a 6 mm FWHM kernel before group-level analysis.

For voxel-wise group analysis, we conducted one-sample $t$ tests comparing the voxel-wise $\Delta S$ values (averaged across both sessions) against zero. We used a lenient threshold of $p < 0.005$, uncorrected (voxel-extend threshold 25) for this test under the assumption that identity expectations ($\Delta S$) might have been disrupted by cTBS in half of the runs. We used an explicit mask containing voxels in the gray matter and the midbrain (see **Univariate test for iPE fMRI signals** above).

We conducted ROI-based pattern similarity analyses by using functional ROIs in the lateral OFC that overlapped with the targeted OFC seed regions. Specifically, we conducted the same analysis described above using activity patterns in clusters of voxels at $p < 0.005$ (uncorrected) around the peak activations (lateral OFC: [−26, 48, −18], [−24, 30, −14]) with at least 25 voxels. For statistical testing, we conducted Wilcoxon signed rank tests. All tests were one-sided, testing our hypothesis that cTBS would disrupt identity expectations.

We also conducted independent $t$ tests (two-sided) and Pearson's correlations exploring if the cTBS effect on identity expectations were related to any between-subject factors such as sex, TMS session order (cTBS or sham first), TMS intensity, and age, but found no significant effects (all $p > 0.1$).

### Regions of interest (ROIs)

This section includes a summary of the ROIs used in our study, including their use in both the small volume correction (SVC) for voxel-wise analysis and in ROI-based statistical testing.

ROIs for the lateral OFC and LPFC were defined based on 8 mm spheres around MNI coordinates (left lateral OFC: [−28, 38, −16]; right lateral OFC: [28, 38, -16]; left LPFC: [-48, 38, 20]; right LPFC: [48, 38, 20]). We refer to these two ROIs as "Targeted OFC seed" and "LPFC stimulation site" respectively. These ROIs were used in the functional connectivity analysis to determine TMS stimulation coordinates (Fig. 1c). They were also used for SVC of the global connectedness voxel-wise analysis (Fig. 2a, b), for SVC of the identity expectation analysis (Fig. 6), and for ROI-based global connectedness analysis across runs and sessions (Fig. 2c).

For the iPE analyses, we defined ROIs in the midbrain, LPFC and OFC as follows. For the LPFC and midbrain (Supplementary Fig 5a), we identified clusters of voxels around the peak activations (left midbrain: [−10, −24, −10]; right midbrain: [10, −14, −10]; left LPFC: [−40, 6, 40]; right LPFC: [46, 22, 30]) that survived whole-brain correction at $p_{FWE} = 0.05$ with a voxel-extend threshold of 25. The functional midbrain ROI was further masked with an anatomical midbrain mask[62] thresholded at 0.5 to exclude voxels that are not part of the midbrain. For the OFC, the ROI (Supplementary Fig 5a) included voxels at $p < 0.001$ (uncorrected) from clusters around peak voxels that survived $p_{FWE-SVC} = 0.05$ for the Targeted OFC seed region (left OFC: [−22, 30, −14]; right OFC: [20, 42, −16]).

For comparing reward identity expectations between cTBS and sham sessions, we defined unbiased ROIs in the OFC based on a searchlight analysis that combined data from both the sham and cTBS session. Specifically, we used a lenient threshold $p = 0.005$ to identity clusters that contained reward identity expectations and defined the ROI based on clusters in the lateral OFC that overlapped with the Targeted OFC seed.

### Reporting summary

Further information on research design is available in the Nature Portfolio Reporting Summary linked to this article.

## Data availability

The behavioral and processed imaging data supporting the findings presented here are available on GitHub (https://github.com/QingfangLiu/OFC_midbrain_TMS). Source data are provided with this paper. Statistical group-level maps are available on NeuroVault (https://neurovault.org/collections/15898/). The probabilistic midbrain atlas is available from the authors of the original article (https://www.adcocklab.org/neuroimaging-tools). Source data are provided with this paper.

## Code availability

Analysis code for reproducing the findings reported in this manuscript is available on GitHub (https://github.com/QingfangLiu/OFC_

midbrain_TMS, https://doi.org/10.5281/zenodo.10537266) with relevant instructions. Any additional information required to reproduce the findings is available from the lead contact upon request.

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

## Acknowledgements

We thank Donnisa Edmonds, Hamna Siddiqui, and Laura Shanahan for their assistance in fMRI data acquisition. We also thank James Howard for help with designing the trans-reinforcer reversal learning task, as well as Phillip Witkowski, Avinash Vaidya, and Yihong Yang for helpful discussions. This work was supported by National Institute on Deafness and Other Communication Disorders grant R01DC015426 (to T.K.) and the Intramural Research Program at the National Institute on Drug Abuse (ZIA DA000642 to T.K and DA000587 to G.S.). The opinions expressed in this work are the authors' own and do not reflect the view of the NIH/DHHS.

## Author contributions

Q.L. and T.K. designed the experiment. Q.L., Y.Z. and S.A. collected the data. Q.L. and T.K. analyzed the data. Q.L., Y.Z., S.A., J.L.V., G.S. and T.K. wrote the manuscript.

## Competing interests

The authors declare no competing interests.
