## [Peer Review File · Nature Communications]

Midbrain signaling of identity prediction errors depends on orbitofrontal cortex networksREVIEWER COMMENTS

Reviewer #1 (Remarks to the Author):

Prior research in rodents has found that interaction between the orbitofrontal cortex (OFC) and neurons in the dopaminergic midbrain is essential for updating stimulus-outcome associations. Whether this is true in humans has, however, been outside the reach of causal manipulation until recently. Here Liu and colleagues use a network-level transcranial magnetic stimulation (TMS) approach to alter activity within the lateral OFC of people performing an odor reversal task. They report that this TMS protocol is able to transiently modulate activity within lateral OFC for a brief period of time after stimulation and during that period, subjects exhibit slight alterations in their choices around reversals of stimulus-odor mappings. They go on to show that degrading lateral OFC activity has a direct effect on identity prediction errors in the dopaminergic midbrain. Thus, the authors provide the first causal evidence in humans that interaction between OFC and midbrain is required for updating stimulus-outcome associations.

This is a clearly written manuscript that describes a set of well-reasoned and executed experiments. The findings make an important contribution to our understanding of how stimulus-outcome associations are updated in the human brain through OFC-dopaminergic circuits. The task is cleverly designed, and the results will be of interest to many in the field of cognitive neuroscience as they confirm and extend prior research that pointed for a direct role for OFC in the formation of prediction errors. Where I have issues, they are relatively minor and related to bolstering the effects that are reported.

1) The TMS approach for modulating activity within the lateral OFC through lateral PFC is ingenious. I have a number of questions and clarifications on the specificity of this approach.

a) How was the location in lateral OFC that would be the target in each subject chosen? I'm assuming that each subject wasn't scanned during the task and a location of increased activity related to outcome expectations determined from that, so why was this particular location chosen? The authors provide two reference for this location ([32,33]) in the main text and a little more detail in the methods but I'm unclear as to what specific features marked this anatomically defined area out as the one to target. Given how important this

target is for the premise of the experiment, I think that the authors should provide a little more information about how the target was selected.

b) I'm interested to know a little bit more about how TMS applied to lateral PFC potentially modulates networks of areas, not just lateral OFC. Figure 2a/b shows that there is a change in connectedness in lateral OFC as well as lateral PFC. Are any other areas in the brain that also show a change in connectedness following TMS to lateral PFC? What I'm wondering here is if the effects of TMS to lateral PFC are truly specific to lateral OFC or if there are other places in the brain that are functionally connected to lateral PFC where there are also effects that could be contributing to altered behavior/prediction errors as reported later in the manuscript.

So, are there other areas in circuits that have been linked to updating valuations by prior research (ACC, amygdala, hippocampus, mediodorsal thalamus etc), or that are modulated by prediction error responses in Figure 5 that show a change in connectedness following TMS to lateral PFC? If yes, and the same analyses as presented in Figures 2c are conducted for these other areas that are modulated, what does the pattern of connectedness look like over time? Is it more like OFC or lateral PFC? Similarly, if the analyses in Figure 5c or 6d are repeated for these other areas that are modulated by TMS what is the result? Depending on the results of this analysis the authors should consider including this either to show how specific their findings are to lateral OFC/PFC and midbrain or to highlight that the effects are related to modulation of an OFC-centered network that is required for stimulus-outcome associations.

c) one of the reasons for the above questions/analysis is that while the present paper does a quite beautiful job of showing that prediction errors are modulated by changes in lateral OFC, there is no clear sense of where the expected and experienced outcomes are being compared in the brain. This is a question that may be beyond the scope of the present manuscript but either an analysis on this point or a short paragraph in the discussion on this point feels like it would enhance the manuscript.

2) Somewhat related to the above, the authors should consider moving what is shown in extended data Figure 3b to the main manuscript as part of Figure 6. The reason for doing this is to show the specificity of the effects to lateral OFC (as lateral PFC doesn't show the

same pattern of effects). This is important to highlight that the effects on expectation are specific to OFC.

Minor comments

- 1) Please include the number of subjects in the results section.
- 2) I may have missed it, but how were the specific odors selected for each subject?
- 3) Figure 2 – the authors should consider adding a bar/violin/box plot of the effect of TMS at the individual subject level for the places where the effects are maximal. At the moment the line plots with SEM don't really give a sense of the range of responses.
- 4) In extended data figure 2, I believe that the authors show that lateral OFC is encoding identity prediction errors and that these are modulated by TMS. Given that prior research by some of this group has failed to find prediction errors in the activity of single neurons in rats, how do the authors interpret this effect? Is it related to dopaminergic inputs modulating the BOLD response, species differences or something else?

Reviewer #2 (Remarks to the Author):

The present study aims to elucidate the causal relationships between the representation of expected outcomes and the encoding of identity prediction errors in the brain.

Understanding this network is of paramount importance for comprehending how the brain effectively assigns appropriate credit during the process of learning and updating internal models or representations in a dynamic environment. To investigate this phenomenon, a combination of fMRI and TMS techniques was employed, allowing for the examination of neural computation alterations related to identity prediction errors and the observation of reduced behavioral flexibility during an identity reversal learning task. Specifically, the study focused on perturbing the identity representation in the lateral OFC to shed light on these intricate cognitive processes.

I appreciate the well-designed task and the results presented in this study, which provides strong support for the causal relationships between the lateral OFC and the dopaminergic midbrain in computing the identity prediction error. I have a few questions and (non-critical)

comments that I believe could be helpful in improving the clarity and presentation of this intriguing finding.

1. Have the authors identified any correlation between the degree to which TMS influences behavioral flexibility or learning rate decreases, and the associated changes in identity expectation within the OFC or alterations in identity prediction error within the midbrain across participants or experimental runs? In simpler terms, does a relationship exist between the effects of TMS on behavioral flexibility and the concurrent neural changes in the OFC and/or midbrain? If a significant correlation is not found, it should not undermine the main argument of this finding. However, if a correlation is indeed present, it would further strengthen the argument.

2. I have some questions regarding the analysis methods utilized in this study. Firstly, how was the representation similarity between rev-1, rev, and rev+1 computed? Was it calculated per reversal point and then averaged? If that is the case, I am interested in knowing the number of trials included in this analysis, which would indicate the number of reversals each participant experienced. This information would help me understand the stability of the representation of expected order identity across the samples. Furthermore, if the representation similarity was computed within the same reversal point, I am curious if there were any effects caused by temporal autocorrelation. To address this potential issue, the authors might consider computing the representational similarity across different reversal points. For instance, an alternative approach could involve measuring the representational similarity between activity patterns of trials in which different visual cues are associated with the anticipation of the same order outcome. This would provide further insights into the robustness of the observed effects and potential temporal dependencies within the data.

3. It would be greatly appreciated if the authors could provide additional details on how the global connectedness was computed. Did the authors employ a general linear model to control for other variables, such as connectedness estimated from white matter? Clarifying this aspect would enhance the understanding of the methodology and the potential confounding factors that were considered in the analysis.

4. Could the authors please include a section in the methods explaining how the regions of interest (ROIs) were defined? It would be helpful to know, for instance, which specific ROI(s) were subjected to small volume correction in the statistical analysis.

5. While I understand the rationale behind the authors' choice of the lateral OFC and LPFC to investigate the effects of cTBS, I also recognize that the same analysis could be conducted in other brain regions where the authors observed identity prediction errors, such as the ACC, mPFC, and Insula. Although the interpretation of these additional results may not be necessary as they extend beyond the primary focus of the study, providing supplementary findings would still assist readers in understanding the broader context and facilitate the design of future studies.

6. A typo in line 274 "... provided a better account of behavioral accuracy than the mode(l) with fixed learning rates ..."

Reviewer #3 (Remarks to the Author):

Liu and colleagues examined the neural mechanisms underlying reward-based decision-making using a combination of transcranial magnetic stimulation (TMS) and functional magnetic resonance imaging (fMRI). The authors applied continuous theta-burst stimulation (cTBS) to a lateral prefrontal cortex (LPFC) area determined to be connected to the orbitofrontal cortex (OFC). cTBS was hypothesized to disrupt OFC processing. cTBS was followed by fMRI during a reversal learning task requiring learning associations among cues and rewarding food odors. LPFC-cTBS was compared to sham. The authors found that relative to sham, cTBS disrupted the global connectedness of the OFC, which was taken as a confirmation of target engagement. This effect was observed early (first run of scanning) but diminished over time. Disruption of global connectedness of LPFC showed similar pattern with disruption lasting a bit longer in time. Behaviorally, relative to sham, LPFC-cTBS resulted in reduced learning rates. These reduced learning rates were mirrored by persistence of hypothetical prediction error-related responses in the midbrain, LPFC, and cingulo-opercular areas. Finally, multi-variate analyses demonstrated that cue-evoked activation patterns in the OFC changed less following cTBS than sham. Collectively, these

data suggest that LPFC-cTBS disrupts the appropriate updating of cue-identity associations following reversals, leading to slower reward learning, and increased prediction errors.

This is an extremely intriguing study with an elegant experimental design, strong theoretical motivation, sophisticated analysis techniques, and a powerful multi-modal combination. The results appear to tell a coherent story about mechanisms underlying reward-based decision-making and further establish the efficacy of LPFC-cTBS to modulate OFC function – a technique which this group has pioneered. That said, I am concerned about the extent to which differences of non-interest between the cTBS and sham conditions confound the results. For a journal of this tier, I believe that additional experimentation is needed to establish that the results are not due to confounds. I elaborate on this concern and other more minor concerns below.

Major Concerns

1) I applaud the authors on their rigorous application and assessment of the efficacy of their sham protocol. Unfortunately, as detailed in the methods, despite best efforts to match sham and cTBS, discomfort and perceived intensity was significantly greater for cTBS than sham. It is notable that no perfect control for cTBS exists. So, unless there is reason to believe that discomfort can reasonably lead to observed differences between cTBS and sham, I typically do not belabor the point. However, in this case, I do think there is cause for concern. First, it is remarkable that the time course of many of the effects does not match the documented time course of cTBS on neural function, at least as assessed by MEPs following cTBS to M1 (e.g. Wischniewski and Schutter, 2015, Brain Stim). Instead of lasting in the 45-60 minute range as has been documented, many of the detailed effects persist for only the first run (~15 minutes). One interpretation of this time course would be the time course of persistent discomfort from cTBS.

How would persistent discomfort impact cognitive function? One possibility is physical – discomfort may cause participants to sniff with less vigor, thereby reducing their ability to identify odors. Additionally (or alternatively), discomfort could lead to reduced attention to

the task in lieu of attentiveness to the discomfort. Both of these possibilities could contribute to reductions in odor identification, reversal learning, and neural correlates of these functions.

I see two ways to address this potential confound. The first is statistical – does a model that includes discomfort and intensity ratings still show significant effects of cTBS vs sham? If so, this provides evidence that variance is explained by cTBS over-and-above the physical effects of cTBS. If not, then I'm afraid additional experimentation is warranted. Perhaps a small sample of individuals could be tested using an active control (e.g. cTBS to an equally uncomfortable posterior area). Replication of the effects in even a small sample of properly controlled individuals would greatly strengthen the claims of the manuscript.

2) The authors perform several processing steps to clean the fMRI data of confounds. However, it is well-documented that effects of motion cannot be perfectly removed. Therefore, it will be useful to report motion as a function of run and stimulation condition. In particular, if such data have any semblance to the stimulation x run interaction on global connectedness, this would undermine a neural locus for such effects.

Minor Concerns

1) Many of the statistics reported in the Results report “V” as a test statistic. I'm not familiar with this statistic. Perhaps a reference or a more conventional test statistic could be used.

2) The authors compare the trial prior to reversals to trials following reversals in some analyses. It is unclear why this is an appropriate metric. Their computational model indicates that the trial following reversals is the critical locus of effect, so statistics that focus on that trial exclusively would seem to be more appropriate.

3) It has been documented that damage to the LPFC produces perseveration tendencies despite negative feedback (e.g. Wisconsin Card Sorting – Milner, 1963). Therefore, LPFC-cTBS may lead to impairments in reversal learning due to direct disruption of the LPFC in addition to, or in lieu of, disruption of the OFC. I believe a straightforward test of this would

be to examine LPFC activation at the time of reversal feedback. Diminished LPFC activation following cTBS would be expected if LPFC-cTBS disrupts control processes that enable switching from irrelevant associations to new ones. Absence of such an effect would offer modest credence towards the idea that the OFC, not the LPFC is what is critical here.

4) It appears that some analyses focus on the period of disruption identified by global connectedness (i.e. first run) and others consider all of the data. It seems that all analyses should be brought into alignment. Either effects are restricted to the first run or they are distributed across the entire scanning session. Waffling between these approaches gives the sense of a lack of rigor.

5) In the methods, the univariate analyses make mention of cue-locked regressors, but I presume prediction error related effects are odor-locked. These regressors and contrasts upon them should be explicitly mentioned. If prediction errors are calculated on cue-locked regressors, a clear rationale for how this can map onto prediction errors would need to be established.

6) It appears that multi-variate analyses used an uncorrected threshold. This is not statistically permissible, certainly not in this age of heightened concern about reproducibility resulting from loose standards of the past.

REVIEWER COMMENTS

Reviewer #1 (Remarks to the Author):

Prior research in rodents has found that interaction between the orbitofrontal cortex (OFC) and neurons in the dopaminergic midbrain is essential for updating stimulus-outcome associations. Whether this is true in humans has, however, been outside the reach of causal manipulation until recently. Here Liu and colleagues use a network-level transcranial magnetic stimulation (TMS) approach to alter activity within the lateral OFC of people performing an odor reversal task. They report that this TMS protocol is able to transiently modulate activity within lateral OFC for a brief period of time after stimulation and during that period, subjects exhibit slight alterations in their choices around reversals of stimulus-odor mappings. They go on to show that degrading lateral OFC activity has a direct effect on identity prediction errors in the dopaminergic midbrain. Thus, the authors provide the first causal evidence in humans that interaction between OFC and midbrain is required for updating stimulus-outcome associations.

This is a clearly written manuscript that describes a set of well-reasoned and executed experiments. The findings make an important contribution to our understanding of how stimulus-outcome associations are updated in the human brain through OFC-dopaminergic circuits. The task is cleverly designed, and the results will be of interest to many in the field of cognitive neuroscience as they confirm and extend prior research that pointed for a direct role for OFC in the formation of prediction errors. Where I have issues, they are relatively minor and related to bolstering the effects that are reported.

We thank the reviewer for their positive comments and constructive suggestions. We have addressed these comments in the revised manuscript, as detailed below.

1) The TMS approach for modulating activity within the lateral OFC through lateral PFC is ingenious. I have a number of questions and clarifications on the specificity of this approach. a) How was the location in lateral OFC that would be the target in each subject chosen? I'm assuming that each subject wasn't scanned during the task and a location of increased activity related to outcome expectations determined from that, so why was this particular location chosen? The authors provide two reference for this location ([32,33]) in the main text and a little more detail in the methods but I'm unclear as to what specific features marked this anatomically defined area out as the one to target. Given how important this target is for the premise of the experiment, I think that the authors should provide a little more information about how the target was selected.

We have provided additional clarification regarding the selection of the stimulation target in the Methods as well as a conceptual overview in the Results. In brief, we selected a location to stimulate in the LPFC of each subject based on it having high resting-state functional connectivity with the lateral OFC, using an approach that took into account group-level (normative) connectivity patterns and then used subject-specific resting-state data to identify a person-specific location of maximal connectivity. We focused on a portion of lateral OFC represented by a Montreal Neurological Institute (MNI) coordinate ($x = 28, y = 38, z = -16$). This MNI coordinate was chosen when we first developed this TMS protocol for indirect targeting of OFC via the LPFC (James D. Howard et al., 2020), based on two key criteria. First, we considered previous findings implicating this location within lateral OFC in representation of expected reward identity (Howard et al., 2015). Second, we used a large normative resting-state dataset ($N = 1,000$ available on neurosynth.org) to identify a location that showed strong functional connectivity with an isolated cluster of LPFC voxels that are able to be targeted by TMS. Thus, we focused on an area of lateral OFC that is relevant to our hypotheses regarding cognition and that is a "good" indirect target in the sense of having robust connectivity to stimulation-accessible locations within LPFC. To generate an MNI coordinate for the left lateral OFC, we simply mirrored the coordinate at the midline ($x = -28, y = 38, z = -16$). As can be seen in **Figure R1.1**, the left and right OFC coordinate each show robust resting-state connectivity with an isolated cluster in the neurosynth.org dataset.

Figure R1.1: Left: functional connectivity map of left OFC seed coordinate: $x = -28$, $y = 38$, $z = -16$ in neurosynth.org; Right: functional connectivity map of right OFC seed coordinate: $x = 28$, $y = 38$, $z = -16$.

We have now expanded our description of the network-targeted method in the Results section (Lines 129-137):

Our TMS approach targeted an area in the lateral OFC, which has previously been shown to represent the identity of expected rewards (Howard et al., 2015; Howard & Kahnt, 2017; Klein-Flugge et al., 2013) and which is functionally connected to an isolated cluster in the lateral prefrontal cortex (LPFC) in a large normative data set (neurosynth.org). For each subject and hemisphere, we then used this lateral OFC area as the seed region in a subject-level resting-state fMRI connectivity analysis and identified an area in LPFC that showed maximal connectivity with the OFC seed (**Fig 1c**). Our previous work has shown that continuous theta burst stimulation (cTBS) over these individually selected stimulation sites modulates activity in the lateral OFC network and disrupts outcome-guided behaviors (J. D. Howard et al., 2020; Wang et al., 2020).

We have also clarified the two considerations influencing the choice of these coordinates in the Methods section (Lines 644-650):

The central-lateral OFC coordinates were selected based on previous research linking activity in lateral OFC to outcome-specific expectations (Howard et al., 2015; Howard & Kahnt, 2017; Klein-Flugge et al., 2013) and strong functional connectivity of this region with the LPFC in a large resting-state dataset ($N = 1000$, neurosynth.org). In other words, we focused on an area of lateral OFC that is relevant to our hypotheses regarding the cognitive function and that is a “good” indirect target in the sense of having robust connectivity to stimulation-accessible locations within LPFC.

b) I’m interested to know a little bit more about how TMS applied to lateral PFC potentially modulates networks of areas, not just lateral OFC. Figure 2a/b shows that there is a change in connectedness in lateral OFC as well as lateral PFC. Are any other areas in the brain that also show a change in connectedness following TMS to lateral PFC? What I’m wondering here is if the effects of TMS to lateral PFC are truly specific to lateral OFC or if there are other places in the brain that are functionally connected to lateral PFC where there are also effects that could be contributing to altered behavior/prediction errors as reported later in the manuscript.

So, are there other areas in circuits that have been linked to updating valuations by prior research (ACC, amygdala, hippocampus, mediodorsal thalamus etc), or that are modulated by prediction error responses in Figure 5 that show a change in connectedness following TMS to lateral PFC? If yes, and the same analyses as presented in Figures 2c are conducted for these other areas that are modulated, what does the pattern of connectedness look like over time? Is it more like OFC or lateral PFC? Similarly, if the analyses in Figure 5c or 6d are repeated for these other areas that are modulated by TMS what is the result? Depending on the results of this analysis the authors should consider including this either to show how specific their findings are to lateral OFC/PFC and midbrain or to highlight that the effects are related to modulation of an OFC-centered network that is required for stimulus-outcome associations.

We initially focused our analyses on OFC and LPFC as we had a priori predictions for these regions based on our hypotheses and stimulation protocol. However, we agree with the reviewer that it could be interesting to investigate TMS effects on other brain regions associated with valuation and stimulus-outcome learning. This is particularly relevant for regions that signal iPEs in our experiment, or similar paradigms.

We first explored if there are brain areas outside our regions of interest that showed a change in connectedness following our network-targeted cTBS procedure. Using a whole-brain corrected threshold ($p_{FWE} < 0.05$), we did not find any regions showing significant TMS effects on global connectedness. Next, we explored the effects of TMS on global connectedness in other brain regions that are generally associated with value updating, namely: insula, midbrain, mPFC (including ACC), striatum, thalamus, and amygdala. These areas were defined as follows. For insula, midbrain, mPFC, striatum, and thalamus, we used significant (whole-brain $p_{FWE} < 0.05$) clusters from the iPE analysis in Figure 5 and Extended Data Table 1. For thalamus (right: [10, -12, 4]; left: [-10, -14, 6]), we used the neuromorphometrics atlas to exclude voxels outside the thalamus. In contrast to our previous findings (Howard & Kahnt, 2018; Suarez et al., 2019), we did not observe iPE responses near hippocampus or amygdala, even with rather lenient thresholds. We therefore created an amygdala mask based on the peak coordinates (left: [-20, -4, -20]; right: [16, -6, -14]) identified previously (Howard & Kahnt, 2018). To explore potential effect of TMS on global connectedness in these areas, we conducted two-way repeated measure ANOVAs with TMS and time as two factors in each area (**Extended Data Fig 3**). We found a main effect of TMS in the midbrain ($F(1, 30) = 5.693, p = 0.024$), and a main effect of time in the thalamus ($F(3.07, 92.22) = 5.004, p = 0.003$). However, these main effects were not qualified by significant interactions, rendering them difficult to interpret. In summary, these additional analyzes show that the effects of cTBS were relatively selective to the targeted circuit. We did not find any main or interaction effects in any of the other four areas (all p 's > 0.05). We included these results in **Extended Data Fig 3** and point to them in the main manuscript (Lines 191-195):

We also tested whether OFC-targeted cTBS modulated global connectedness in additional brain areas. First, we ran a whole brain analysis ($p_{FWE} < 0.05$, whole-brain FWE corrected), which did not reveal any significant effects of TMS. Second, we explored effects of TMS on connectedness in several other brain areas commonly associated with reward learning (**Extended Data Fig 3**).

Extended Data Fig 3. Global connectedness in additional brain areas.

We explored the effects of TMS on global connectedness in additional brain regions involved in value updating, namely: insula, midbrain, mPFC (including ACC), striatum, thalamus, and amygdala. These ROIs were defined based on our iPE analysis (insula, midbrain, mPFC, striatum, and thalamus) or based on coordinates identified previously (Howard & Kahnt, 2018) (amygdala). Two-way repeated measure ANOVAs with TMS and time as factors for each area showed a main effect of TMS in the midbrain ($F(1, 30) = 5.693, p = 0.024$), and a main effect of time in the thalamus ($F(3.07, 92.22) = 5.004, p = 0.003$). However, these main effects were not qualified by significant interactions. No main or interaction effects were identified in any of the other four areas (all p 's > 0.05).

We also tested if cTBS modulated iPE responses in these areas (insula, mPFC, striatum, thalamus, and amygdala, **Extended Data Fig 6**). Similar to the analysis reported in **Fig 5c**, we compared the change in outcome-related fMRI responses from rev to rev+1 trials between cTBS and sham, using one-sided paired t-tests. This analysis yielded only one significant effect in the mPFC ROI ($t(30) = 1.94, p = 0.031$), whereas cTBS effects in all other areas (insula, striatum, thalamus, and amygdala) were not significant (all p 's > 0.05). However, the cTBS effect on mPFC activity was not significant on either rev or rev+1 trials (all p 's > 0.05), making it difficult to interpret. Overall, these analyses suggest that the cTBS effect on iPE responses were restricted to the midbrain (**Fig 5c**), LPFC, left OFC (**Extended Data Fig 5**), and mPFC (**Extended Data Fig 6**).

Extended Data Fig 6: Effects of TMS on iPE-related activity in additional brain areas.

(a) Outcome-related fMRI responses on trials before (rev₋₁), during (rev) and following a reversal (rev₊₁ and rev₊₂). (b) Change from rev to rev₊₁ in the sham and cTBS session. ROIs were defined as described in **Extended Data Fig 3**. Note, amygdala responses did not display the typical iPE pattern because this area was not functionally defined in the same way as the other areas. * denotes $p < 0.05$ and n.s. denotes $p > 0.05$.

Following the suggestion by the reviewer, we also tested whether TMS disrupted identity expectations in brain regions other than the lateral OFC. For this, we created functional ROIs based on the peak coordinates from the searchlight analysis in the original **Extended Data Table 2**, including the dorsal striatum ([-20, 22, 4]; [30, 14, -2]), ventral striatum ([18, 12, -14]), midbrain ([-16, -18, -6]; [12, -22, -12]), mPFC ([-4, 26, 46]; [-2, 6, 44]), insula ([-38, 10, -2]; [44, -8, 14]), hippocampus ([-22, -10, -30]), thalamus ([-10, -26, 2]; [6, -8, 8]), and Acc ([-4, 32, 2]; [10, 20, -8]). We repeated the ROI-based multivariate pattern similarity analysis shown in Figure 6d and compared cTBS with sham sessions using one-sided Wilcoxon signed rank tests. We did not observe any significant decreases in reward identity expectations in any ROI (all p 's > 0.05, **Figure R1.2**).

However, these ROIs were created using a lenient threshold ($p = 0.005$, uncorrected) and are not based on any a priori experimental hypothesis. In response to reviewer #3, who noted that uncorrected results should not be included, we have removed **Extended Data Table 2** from the manuscript. Because the analyses in Figure R1.2 are based on these regions, we decided not to include **Figure R1.2** in the revised manuscript.

Figure R1.2: TMS effects on reward identity expectations in other areas (Dorsal Striatum, Ventral Striatum, Midbrain, mPFC, Insula, Hippocampus, Thalamus, Acc). No significant decreases in reward identity expectations were observed in any ROI (all p 's > 0.05).

c) one of the reasons for the above questions/analysis is that while the present paper does a quite beautiful job of showing that prediction errors are modulated by changes in lateral OFC, there is no clear sense of where the expected and experienced outcomes are being compared in the brain. This is a question that may be beyond the scope of the present manuscript but either an analysis on this point or a short paragraph in the discussion on this point feels like it would enhance the manuscript.

We thank the reviewer for prompting us to consider the important question. We agree that our experiment and analyses cannot speak to where in the brain iPEs are computed. However, we believe that the same studies that support the computation of value PEs by midbrain dopamine neurons would also support the computation of iPEs, since these informational streams typically are from brain regions known to signal information about sensory features in addition to value. In particular, the OFC interacts with midbrain dopamine neurons both directly and indirectly to elicit both excitatory and inhibitory responses, and the OFC is well-established to convey information about both the value and identity of expected outcomes. With that said, we added a paragraph in the Discussion section (Lines 455-470) to express our thoughts on this question:

Our results leave open the question whether iPEs are computed within the midbrain or in upstream regions. To compute prediction errors, an area needs information about what is expected and what is received. Dopamine neurons in the VTA have been proposed as a candidate for computing reward prediction errors (Watabe-Uchida et al., 2017), and the same inputs and mechanisms would put dopamine neurons in a good position to compute iPEs. For example, previous studies show that the lateral OFC provides excitatory inputs to GABA-ergic neurons in the VTA (Watabe-Uchida et al., 2012), which, in turn, inhibit VTA dopamine neurons (Faget et al., 2016). Furthermore, dopamine neurons receive direct excitatory input about received rewards from sensory cortex and the lateral hypothalamus (Geisler et al., 2007; Wickersham et al., 2007). While these pathways are typically hypothesized to carry information about value, the upstream regions are known to also encode sensory features of outcomes (Boorman et al., 2016; Delamater, 2007; Howard et al., 2015; Howard & Kahnt, 2017, 2018; Mizrak et al., 2019; Pauli et al., 2019; Rudebeck & Murray, 2014; Stalnaker et al., 2014), which could be used to signal iPEs. Our results showing that modulating activity in lateral OFC changes iPE responses in the midbrain are

compatible with the idea that information about expected reward identity from the lateral OFC converges on dopamine neurons with information about received reward to compute identity prediction errors. However, we cannot rule out the possibility that other, additional regions are involved in the computation of iPEs.

2) Somewhat related to the above, the authors should consider moving what is shown in extended data Figure 3b to the main manuscript as part of Figure 6. The reason for doing this is to show the specificity of the effects to lateral OFC (as lateral PFC doesn't show the same pattern of effects). This is important to highlight that the effects on expectation are specific to OFC.

We thank and agree with the reviewer's comment. However, in response to reviewer #3's minor comment 6, we have decided to only consider the effects of TMS on identity expectations in brain areas that survive a corrected statistical threshold. Whereas identity expectations in lateral OFC survive correction (when using data from the sham sessions only), signals in LPFC do not. As LPFC does not show robust identity expectations, we removed the analysis testing the effects of TMS on identity expectation in this area. Text deleted in the revised manuscript:

~~To test for similar cTBS effects in the LPFC, we also performed the same analysis using the LPFC clusters ([32, 40, 24], $t = 4.35$) that overlapped with the LPFC stimulation site (**Extended Data Fig 3a**). Reward identity expectations in the LPFC (**Extended Data Fig 3b**) were significant in both the sham ($V = 372$, $p = 0.014$, one-sided) and cTBS session ($V = 393$, $p = 0.0036$, one-sided), and there was no significant difference between sessions ($V = 223$, $p = 0.69$, one-sided).~~

~~We also observed identity expectations in the LPFC, consistent with previous research implicating the LPFC in representing task states (Tomov et al., 2018; Tomov et al., 2023; Vaidya et al., 2021). However, in contrast to representations in the lateral OFC, cTBS did not disrupt representations in the LPFC. This suggests that identity expectations in LPFC were not affected by cTBS and thus unlikely to be the cause for the observed behavioral and iPEs effects.~~

Minor comments

1) Please include the number of subjects in the results section.

We added the number of subjects (N=31) to line 88 in the Results section.

2) I may have missed it, but how were the specific odors selected for each subject?

The specific odors for each subject were selected by (1) asking subjects to provide pleasantness ratings for an initial set of ten odors and (2) selecting three odors that were rated as pleasant (above neutral) and most closely matched. The specific procedures have been described in detail in lines 548-555 and Extended Data Fig 1.

3) Figure 2 – the authors should consider adding a bar/violin/box plot of the effect of TMS at the individual subject level for the places where the effects are maximal. At the moment the line plots with SEM don't really give a sense of the range of responses.

We thank the reviewer for this suggestion. To illustrate the range of responses, we created an additional plot showing the difference in global connectedness (cTBS – sham) in individual subjects. For this, we created functional ROIs of clusters in OFC and LPFC based on the voxel-wise comparison between sham and cTBS, shown in Fig 2a, b (thresholded at $p < 0.001$ uncorrected). It is important to note that these ROIs were defined by testing for the TMS effect, and thus these plots are purely illustrative and the effect of TMS should not be interpreted. Each line in the plot depicts the change of global connectedness (cTBS – sham) within each subject. We have included this figure as the **Extended Data Fig 2** and directed our reader to this figure in our main text.

Extended Data Fig 2: Global connectedness (cTBS – sham) in functional ROIs of OFC and LPFC.

Global connectedness difference values (cTBS – sham) within functional ROIs in OFC and LPFC that are defined based on the voxel-wise comparison between sham and cTBS, shown in **Fig 2a, b** (thresholded at $p < 0.001$, uncorrected). Note that these ROIs were defined based on the effect of TMS, and thus plots are purely illustrative. Bold black line depicts the mean and shaded area the standard error across subjects. Thin gray lines are data from individual subjects.

4) In extended data figure 2, I believe that the authors show that lateral OFC is encoding identity prediction errors and that these are modulated by TMS. Given that prior research by some of this group has failed to find prediction errors in the activity of single neurons in rats, how do the authors interpret this effect? Is it related to dopaminergic inputs modulating the BOLD response, species differences or something else?

The reviewer is correct that our current results (and those from other fMRI studies, e.g., (Boorman et al., 2016; Howard & Kahnt, 2018; Suarez et al., 2019)) differ from what has been observed using electrophysiological recordings in rats (Stalnaker et al. (2018)). We believe that several factors may contribute to this difference. First, as the reviewer pointed out, it is possible that species differences between the OFC of rats and primates could play a role (Rudebeck & Izquierdo, 2022). Second, studies in rats typically involve considerably more training compared to studies in humans, where only a brief period of practice precedes the main experiment. These varying levels of training could affect how iPEs are represented in the OFC. Third, as the reviewer suggests there are significant differences between single unit firing rates and BOLD responses, which are more closely related to local field potentials (reflecting presynaptic activity) than multi-unit responses (Logothetis et al., 2001). Thus, it is possible that iPE responses in the OFC as measured with fMRI reflect the influence of input from dopaminergic neurons in the midbrain on neural activity in neural populations that are poorly sampled by unit recording approaches, such as local interneurons, or on subthreshold events that are not visible in spiking activity. We have added a discussion of these discrepancies to the manuscript (Lines 472-484):

In addition to the midbrain, we found identity prediction error responses in other areas, including the striatum, mPFC, thalamus, insula, and lateral OFC. Responses to identity errors in the lateral OFC are consistent with previous human fMRI studies (Boorman et al., 2016; Howard & Kahnt, 2018; Suarez et al., 2019) but contrast with what has been observed using electrophysiological recordings in rats (Stalnaker et al., 2018). This discrepancy could reflect differences between species or training requirements but seems most likely to be due to the different neural recoding methods. Specifically, single-unit recording approaches measure spiking activity

and are biased to sample from large pyramidal neurons, whereas BOLD responses are more closely related to local field potentials (reflecting presynaptic activity) than multi-unit responses (Logothetis et al., 2001). This suggests that fMRI responses to iPEs in the lateral OFC could reflect the influence of input from dopaminergic neurons on activity in neural populations that are poorly sampled by unit recording approaches (e.g., local interneurons) or on subthreshold events that are not visible in spiking activity.

Reviewer #2 (Remarks to the Author):

The present study aims to elucidate the causal relationships between the representation of expected outcomes and the encoding of identity prediction errors in the brain. Understanding this network is of paramount importance for comprehending how the brain effectively assigns appropriate credit during the process of learning and updating internal models or representations in a dynamic environment. To investigate this phenomenon, a combination of fMRI and TMS techniques was employed, allowing for the examination of neural computation alterations related to identity prediction errors and the observation of reduced behavioral flexibility during an identity reversal learning task. Specifically, the study focused on perturbing the identity representation in the lateral OFC to shed light on these intricate cognitive processes.

I appreciate the well-designed task and the results presented in this study, which provides strong support for the causal relationships between the lateral OFC and the dopaminergic midbrain in computing the identity prediction error. I have a few questions and (non-critical) comments that I believe could be helpful in improving the clarity and presentation of this intriguing finding.

We thank the reviewer for their encouraging and constructive comments. Below we outline how we have addressed these points in the revised manuscript.

1. Have the authors identified any correlation between the degree to which TMS influences behavioral flexibility or learning rate decreases, and the associated changes in identity expectation within the OFC or alterations in identity prediction error within the midbrain across participants or experimental runs? In simpler terms, does a relationship exist between the effects of TMS on behavioral flexibility and the concurrent neural changes in the OFC and/or midbrain? If a significant correlation is not found, it should not undermine the main argument of this finding. However, if a correlation is indeed present, it would further strengthen the argument.

We thank the reviewer for suggesting testing correlations between behavioral and neural effects of TMS. In response, we tested if the behavioral effect of TMS (difference in the change in outcome prediction accuracy from pre- to post-reversal between the cTBS and sham session) was correlated with effects of TMS on (1) iPE responses in the midbrain (difference in the change of outcome-related responses from reversal to post-reversal trials between the cTBS and sham session) and (2) reward identity expectations in the OFC (difference in delta S between the cTBS and sham session). However, these two correlations were not significant (1: $r = 0.154$, $p = 0.41$; 2: $r = -0.105$, $p = 0.58$). In the revised manuscript, we now describe these correlation results in the Results section.

Lines 334-336: Moreover, there was no significant correlation between the behavioral and neural effect of cTBS in the midbrain (\$r = 0.154\$, \$p = 0.41\$ ).

Lines 421-422: However, the cTBS effect on identity expectation in the lateral OFC was not significantly correlated with the behavioral effect of cTBS (\$r = -0.105\$, \$p = 0.58\$ ).

2. I have some questions regarding the analysis methods utilized in this study. Firstly, how was the representation similarity between rev-1, rev, and rev+1 computed? Was it calculated per reversal point and then averaged? If that is the case, I am interested in knowing the number of trials included in this analysis, which would indicate the number of reversals each participant experienced. This information would help me understand the stability of the representation of expected order identity across the samples. Furthermore, if the representation similarity was computed within the same reversal point, I am curious if there were any effects caused by temporal autocorrelation. To address this potential issue, the authors might consider computing the representational similarity across different reversal points. For instance, an alternative approach could involve measuring the representational similarity between activity patterns of trials in which different visual cues are associated with the anticipation of the same odor outcome. This would provide further insights into the robustness of the observed effects and potential temporal dependencies within the data.

The reviewer is correct that the pattern similarity between rev-1, rev, and rev+1 was computed per reversal point and then averaged across reversals. For this, we included all trials from the experiment irrespective of subjects' behavioral response. The total number of reversals per session was 36 (i.e., 6 reversals per cue x 2 cues x 3 runs). We clarified this point on lines 978-980:

In this analysis, we included all trials from the experiment irrespective of subjects' behavioral response. The total number of reversals per session was 36 (i.e., 6 reversals per cue x 2 cues x 3 runs).

We fully understand the reviewer's concern regarding a potential effect of temporal autocorrelation and in fact considered this point when planning our analysis. However, we believe it is not a problem. Specifically, the random interleaving of the two cues across trials, combined with our long trial duration (ranging from 12-14s), helps to mitigate potential concerns regarding temporal autocorrelation. That is, rev-1 and rev trials, as well as rev and rev+1 trials are not necessarily subsequent but are typically separated by trials in which the other cue is presented. Across reversals, the average time between rev-1 and rev trials was 27.0 ± 2.3 s for sham, 26.7 ± 2.3 s for cTBS sessions, and the average time between rev and rev+1 trials was 25.5 ± 2.2 s for sham, 25.8 ± 2.3 s for cTBS. A two-way repeated measure of ANOVA (factors pre vs. post reversal and cTBS vs sham) did not yield any significant main effect or interaction (all p's > 0.1). As such, while we cannot fully exclude the possibility of temporal autocorrelation between those trials, there is no compelling basis to assume that such autocorrelation would differ between the rev-1 and rev trials and the rev and rev+1 trials. Importantly, because these potential effects of autocorrelation should be identical between different TMS sessions, there is also no reason to believe these effects could explain differences between cTBS and sham.

We appreciate the suggested analysis but there are several reasons for why this analysis may not work well in our situation. First, previous studies have indicated that the OFC encodes both information about reward identity (generalizing across cue) and cue-specific reward identity information (Klein-Flugge et al., 2013). Therefore, computing pattern similarity using neural responses associated with different cues may inadvertently distort the neural representation space for identity expectation. Second, and more importantly, there is considerable variability in the identity expectations across different reversal trials, and conducting the analysis across reversal points would cause misalignment between different trials. In contrast, by calculating representation similarity from trials surrounding the same reversal, we ensure that the change in the strength of outcome identity expectations is accurately determined by directly comparing the pre-reversal (S_{same}) with its corresponding post-reversal similarity measure ($S_{different}$).

Nevertheless, to fully address this comment, we conducted the alternative analysis approach suggested by the reviewer within the OFC ROI identified in our original analysis. We calculated pattern correlations using reversals associated with the same odor outcomes (pre and post reversal) but with different cues. Specifically, we first categorized six unique reversal types based on the pre-reversal odor outcome and post-reversal odor outcome (i.e., odor 1 to 2, odor 1 to 3, odor 2 to 1, odor 2 to 3, odor 3 to 1, odor 3 to 2), for each cue. Next, we obtained the neural responses for each cue at each of the rev-1, rev, and rev+1 trials. Then for each reversal type, we computed the pattern similarity between rev-1 and rev, and between rev and rev+1, each using neural responses from different cues. Finally, we computed the identity expectation by subtracting the two Fisher z-transformed correlation values and averaged across reversal trials. However, this analysis did not yield significant identity representations across both sessions ($p = 0.55$) and no differences between cTBS and sham ($p = 0.32$) in the OFC.

3. It would be greatly appreciated if the authors could provide additional details on how the global connectedness was computed. Did the authors employ a general linear model to control for other variables, such as connectedness estimated from white matter? Clarifying this aspect would enhance the understanding of the methodology and the potential confounding factors that were considered in the analysis.

The reviewer is correct that we employed a general linear model to control for other variables (motion, white-matter signal, etc.). We realize that our original description of this analysis may not have provided sufficient clarity for others to replicate it. To improve that, we have revised the Method section (see Lines 851-865):

We regressed out all above regressors through a linear filtering process. We denote the number of volumes involved in the analysis as n_{vol} , the number of gray matter voxels in 3mm space as n_{vox} , and the number of above regressors as n_{reg} . We created a filter matrix F containing the above regressors of dimension $(n_{vol} \times n_{reg})$, where all regressors except for the constant were z-scored. We also organized fMRI responses of each gray matter voxel in a matrix B of dimension $(n_{vol} \times n_{vox})$. Then the weight W can be obtained from the least square solution of a linear regression that predicts B using F as

$$W = (F'F)^{-1}(F'B),$$

where $(.)'$ denotes transpose of a matrix and $(.)^{-1}$ denotes the inverse of a matrix. Global connectedness was calculated based on the residual matrix R after removing the variance explained by the filter matrix F from the fMRI responses, where

$$R = B - FW.$$

4. Could the authors please include a section in the methods explaining how the regions of interest (ROIs) were defined? It would be helpful to know, for instance, which specific ROI(s) were subjected to small volume correction in the statistical analysis.

We fully agree with the reviewer's suggestion that having a dedicated section to describe how the ROIs were defined would enhance the clarity of the manuscript. We added a new section titled "Regions of interest (ROIs)" section at the end of the methods section (lines 1007-1035). In this section, we provide a summary of the ROIs used in our study, including their use in both the small volume correction (SVC) for voxel-wise analysis and in ROI-based statistical testing. We also direct the reader to the corresponding figures in the manuscript for each ROI. We retained descriptions of the ROIs in each specific analysis section to facilitate quick access of ROI-related information.

Regions of interest (ROIs)

This section includes a summary of the ROIs used in our study, including their use in both the small volume correction (SVC) for voxel-wise analysis and in ROI-based statistical testing.

ROIs for the lateral OFC and LPFC were defined based on 8 mm spheres around MNI coordinates (left lateral OFC: [-28, 38, -16]; right lateral OFC: [28, 38, -16]; left LPFC: [-48, 38, 20]; right LPFC: [48, 38, 20]). We refer to these two ROIs as "Targeted OFC seed" and "LPFC stimulation site" respectively. These ROIs were used in the functional connectivity analysis to determine TMS stimulation coordinates (**Fig 1c**). They were also used for SVC of the global connectedness voxel-wise analysis (**Fig 2a, b**), for SVC of the identity expectation analysis (**Fig 6**), and for ROI-based global connectedness analysis across runs and sessions (**Fig 2c**).

For the iPE analyses, we defined ROIs in the midbrain, LPFC and OFC as follows. For the LPFC and midbrain (**Extended Data Fig 5a**), we identified clusters of voxels around the peak activations (left midbrain: [-10, -24, -10]; right midbrain: [10, -14, -10]; left LPFC: [-40, 6, 40]; right LPFC: [46, 22, 30]) that survived whole-brain correction at $p_{FWE} = 0.05$ with a voxel-extend threshold of 25. The functional midbrain ROI was further masked with an anatomical midbrain mask (Murty et al., 2014) thresholded at 0.5 to exclude voxels that are not part of the midbrain. For the OFC, the ROI (**Extended Data Fig 5a**) included voxels at $p < 0.001$ (uncorrected) from clusters around peak voxels that survived $p_{FWE-SVC} = 0.05$ for the Targeted OFC seed region (left OFC: [-22, 30, -14]; right OFC: [20, 42, -16]).

For comparing reward identity expectations between cTBS and sham sessions, we defined unbiased ROIs in the OFC based on a searchlight analysis that combined data from both the sham and cTBS session. Specifically, we used a lenient threshold $p = 0.005$ to identify clusters that contained reward identity expectations and defined the ROI based on clusters in the lateral OFC that overlapped with the Targeted OFC seed.

5. While I understand the rationale behind the authors' choice of the lateral OFC and LPFC to investigate the effects of cTBS, I also recognize that the same analysis could be conducted in other brain regions where the authors observed identity prediction errors, such as the ACC, mPFC, and Insula. Although the interpretation of these additional results may not be necessary as they extend beyond the primary focus of the study, providing supplementary findings would still assist readers in understanding the broader context and facilitate the design of future studies.

We agree with the reviewer that conducting additional exploratory analyses outside of our a priori regions of interest may help readers understand the broader context and facilitate future studies. In fact, Reviewer 1 had a similar comment. We therefore also tested the effects of cTBS on global connectedness and iPE responses in additional ROIs, including insula, midbrain, mPFC (including the ACC), striatum, thalamus, and amygdala using two-way repeated measure ANOVAs. We found a main effect of TMS in the midbrain ($F(1, 30) = 5.693, p = 0.024$), and a main effect of time in the thalamus ($F(3.07, 92.22) = 5.004, p = 0.003$). However, these main effects were not qualified by significant interactions, rendering them difficult to interpret. We did not find any main or interaction effects in any of the other four areas (all p 's > 0.05). In summary, these additional analyses show that the effects of cTBS were relatively selective to the targeted circuit. We included these results in **Extended Data Fig 3** and point to them in the main manuscript (Lines 191-195).

We also tested whether OFC-targeted cTBS modulated global connectedness in additional brain areas. First, we ran a whole brain analysis ($p_{FWE} < 0.05$, whole-brain FWE corrected), which did not reveal any significant effects of TMS. Second, we explored effects of TMS on connectedness in several other brain areas commonly associated with reward learning (**Extended Data Fig 3**).

Extended Data Fig 3. Global connectedness in additional brain areas.

We explored the effects of TMS on global connectedness in additional brain regions involved in value updating, namely: insula, midbrain, mPFC (including ACC), striatum, thalamus, and amygdala. These ROIs were defined based on our iPE analysis (insula, midbrain, mPFC, striatum, and thalamus) or based on coordinates identified previously (Howard & Kahnt, 2018) (amygdala). Two-way repeated measure ANOVAs with TMS and time as factors for each area showed a main effect of TMS in the midbrain ($F(1, 30) = 5.693, p = 0.024$), and a main effect of time in the thalamus ($F(3.07, 92.22) = 5.004, p = 0.003$). However, these main effects were not qualified by significant interactions. No main or interaction effects were identified in any of the other four areas (all p 's > 0.05).

We also tested if cTBS modulated iPE responses in these areas. Similar to the analysis reported in **Fig 5c**, we compared the change in outcome-related fMRI responses from rev to rev+1 trials between cTBS and sham, using one-sided paired t-tests. This analysis yielded only one significant effect in the mPFC ROI ($t(30) = 1.94$, $p = 0.031$), whereas cTBS effects in all other areas (insula, striatum, thalamus, and amygdala) were not significant (all p 's > 0.05). However, the cTBS effect on mPFC activity was not significant on either rev or rev+1 trials (all p 's > 0.05), making it difficult to interpret. Overall, these analyses suggest that the cTBS effects on iPE responses outside our targeted areas were only found mPFC, These results are now included in **Extended Data Fig 6**.

Extended Data Fig 6: Effects of TMS on iPE-related activity in additional brain areas.

(a) Outcome-related fMRI responses on trials before (rev₋₁), during (rev) and following a reversal (rev₊₁ and rev₊₂). (b) Change from rev to rev₊₁ in the sham and cTBS session. ROIs were defined as described in **Extended Data Fig 3**. Note, amygdala responses did not display the typical iPE pattern because this area was not functionally defined in the same way as the other areas. * denotes $p < 0.05$ and n.s. denotes $p > 0.05$.

6. A typo in line 274 “... provided a better account of behavioral accuracy than the mode(l) with fixed learning rates ...”

We thank the reviewer for pointing this out. This typo has now been fixed in the manuscript.

Reviewer #3 (Remarks to the Author):

Liu and colleagues examined the neural mechanisms underlying reward-based decision-making using a combination of transcranial magnetic stimulation (TMS) and functional magnetic resonance imaging (fMRI). The authors applied continuous theta-burst stimulation (cTBS) to a lateral prefrontal cortex (LPFC) area determined to be connected to the orbitofrontal cortex (OFC). cTBS was hypothesized to disrupt OFC processing. cTBS was followed by fMRI during a reversal learning task requiring learning associations among cues and rewarding food odors. LPFC-cTBS was compared to sham. The authors found that relative to sham, cTBS disrupted the global connectedness of the OFC, which was taken as a confirmation of target engagement. This effect was observed early (first run of scanning) but diminished over time. Disruption of global connectedness of LPFC showed similar pattern with disruption lasting a bit longer in time. Behaviorally, relative to sham, LPFC-cTBS resulted in reduced learning rates. These reduced learning rates were mirrored by persistence of hypothetic prediction error-related responses in the midbrain, LPFC, and cingulo-opercular areas. Finally, multi-variate analyses demonstrated that cue-evoked activation patterns in the OFC changed less following cTBS than sham. Collectively, these data suggest that LPFC-cTBS disrupts the appropriate updating of cue-identity associations following reversals, leading to slower reward learning, and increased prediction errors.

This is an extremely intriguing study with an elegant experimental design, strong theoretical motivation, sophisticated analysis techniques, and a powerful multi-modal combination. The results appear to tell a coherent story about mechanisms underlying reward-based decision-making and further establish the efficacy of LPFC-cTBS to modulate OFC function – a technique which this group has pioneered. That said, I am concerned about the extent to which differences of non-interest between the cTBS and sham conditions confound the results. For a journal of this tier, I believe that additional experimentation is needed to establish that the results are not due to confounds. I elaborate on this concern and other more minor concerns below.

We thank the reviewer for their encouraging and insightful feedback. Below we outline how we have addressed their concerns in the revised manuscript.

Major Concerns

1) I applaud the authors on their rigorous application and assessment of the efficacy of their sham protocol. Unfortunately, as detailed in the methods, despite best efforts to match sham and cTBS, discomfort and perceived intensity was significantly greater for cTBS than sham. It is notable that no perfect control for cTBS exists. So, unless there is reason to believe that discomfort can reasonably lead to observed differences between cTBS and sham, I typically do not belabor the point. However, in this case, I do think there is cause for concern. First, it is remarkable that the time course of many of the effects does not match the documented time course of cTBS on neural function, at least as assessed by MEPs following cTBS to M1 (e.g. Wischnewski and Schutter, 2015, Brain Stim). Instead of lasting in the 45-60 minute range as has been documented, many of the detailed effects persist for only the first run (~15 minutes). One interpretation of this time course would be the time course of persistent discomfort from cTBS.

How would persistent discomfort impact cognitive function? One possibility is physical – discomfort may cause participants to sniff with less vigor, thereby reducing their ability to identify odors. Additionally (or alternatively), discomfort could lead to reduced attention to the task in lieu of attentiveness to the discomfort. Both of these possibilities could contribute to reductions in odor identification, reversal learning, and neural correlates of these functions.

I see two ways to address this potential confound. The first is statistical – does a model that includes discomfort and intensity ratings still show significant effects of cTBS vs sham? If so, this provides evidence that variance is explained by cTBS over-and-above the physical effects of cTBS. If not, then I'm afraid additional experimentation is warranted. Perhaps a small sample of individuals could be tested using an active control (e.g. cTBS to an equally uncomfortable posterior area). Replication of the effects in even a small sample of properly controlled individuals would greatly strengthen the claims of the manuscript.

We agree with the reviewer that physical discomfort may be a potential confounding factor, and we are thankful for the suggestions on how to address this issue. To address this concern statistically, we included the self-reported discomfort and perceived TMS intensity as predictors into linear mixed effect models for each analysis presented in Fig 2c, 4b, 5c, 6d, and tested whether adding TMS as a predictor significantly improved the model fit (after accounting for the effects of the two control variables). Specifically, we compared the model fits of two linear mixed effect models: a full model that included the effects of subjects, TMS session, discomfort, and intensity as predictors, and a reduced model without the TMS session effect. Additionally, for the voxel-wise Global connectedness analysis, we added the subject-wise difference (sham - cTBS) of the two TMS ratings as a covariate to the group-level one-sample t-test.

In brief, all these analyses replicated our original results, demonstrating that discomfort and perceived TMS intensity cannot account for our results. The results of these analyses are described in more detail below.

- To test if the voxel-wise results of global connectedness were driven by subjects' physical experience, we included discomfort and perceived TMS intensity differences (sham – cTBS) as covariates in the one-sample t-test, which yielded almost identical maps as those shown in Fig 2a, b.
- To test if the TMS effects on global connectedness in the OFC and LPFC ROIs remained significant after accounting for potential confounding factors of self-reported discomfort and perceived TMS intensity, we compared the model fits of two linear mixed effect models on predicting global connectedness for both the OFC (focused on time bins 1 and 2) and LPFC (focused on time bins 2, 3, and 4) effects. Adding TMS session as a predictor significantly improved the model fit for most time bins (all p's <0.05), except for the 3rd time bin in the LPFC (p = 0.183).
- To test if TMS modulated the change in accuracy from pre-reversal to post-reversal after accounting for potential confounding factors of self-reported discomfort and perceived TMS intensity, we compared the fits of two linear mixed effect models. Adding TMS session as a predictor significantly improved the model fit (p = 0.011).
- To test if TMS modulated the iPE-related fMRI responses after accounting for potential confounding factors of self-reported discomfort and perceived TMS intensity, we compared the fits of two linear mixed effect models that predict the change of iPE signals from reversal to post-reversal trials. This analysis was conducted separately for each ROI (midbrain, LPFC, and left OFC). Adding TMS session as a predictor significantly improved the model fit in all ROIs (all p's < 0.05).
- To test if TMS effects on neural representations of expected reward identity in the lateral OFC persisted after accounting for potential confounding effects of self-reported discomfort and perceived TMS intensity, we compared model fits of two linear mixed effect models. Adding TMS session as a predictor significantly improved the model fit for the effect in the lateral OFC (p = 0.043).

Taken together, we conclude that our results are not driven by discomfort and perceived TMS intensity. We added a summary of these control analyses (lines 635-639) in the revised manuscript.

For each of the analyses presented in the manuscript, we considered discomfort and perceived TMS intensity as potential confounding variables and performed control analyses to test whether they could account for the results. All control analyses show that the TMS effects reported here cannot be explained by subjects' physical discomfort or perceived TMS intensity.

In line with reviewer's suggestion, we feel that the results from these control analyses provide evidence that variance is explained by cTBS over-and-above the physical effects of cTBS and thus that the additional control experiment is not necessary.

2) The authors perform several processing steps to clean the fMRI data of confounds. However, it is well-documented that effects of motion cannot be perfectly removed. Therefore, it will be useful to report motion as a function of run and stimulation condition. In particular, if such data have any semblance to the stimulation x run interaction on global connectedness, this would undermine a neural locus for such effects.

We agree that effects of head motion cannot be entirely removed and that it is necessary to examine how motion changes across TMS sessions and runs. Therefore, we further analyzed the six motion parameters (three translation parameters and three rotation parameters) estimated during the realignment. Specifically, we calculated the sum of absolute changes in each motion parameter on a volume-by-volume basis within each half of each run (averaging across the three translation and three rotation parameters) to bring them on the same time scale as the global connectedness analysis.

We first conducted a set of two-way repeated measure ANOVAs to examine the effect of TMS and time on both translation and rotation parameters. For translation, this did not reveal any significant main effect of TMS or time, or their interaction (all p 's > 0.05). For rotation, we observed a significant main effect of time ($p = 0.010$), but there was no main effect of TMS or a time by TMS interaction (all p 's > 0.05).

We further tested whether the effect of TMS on the global connectedness in the lateral OFC and LPFC survive when accounting for the effects of head motion. To do this, we compared the fits of two linear mixed effect models: a full model that included effects from subjects, TMS session, time bin, two motion parameters (average translation parameter and average rotation parameter) as predictors, and a reduced model without the TMS session effect. This analysis was conducted for both the OFC (focused on time bins 1 and 2) and LPFC (focused on time bins 2, 3, and 4) effects. Adding TMS session as a predictor significantly improved the model fit for both the OFC (time bin 1: $p = 0.0011$; time bin 2: $p = 0.0069$) and LPFC (time bin 2: $p = 0.00063$; time bin 3: 0.0065; time bin 4: $p = 6.069e-5$), demonstrating that the effect of TMS on global connectedness was not driven by differences in head motion. We included these results as **Extended Data Fig 4**.

Extended Data Fig 4: Comparison of head motion parameters across runs between cTBS and sham sessions. (a) Translation and (b) rotation parameters during scanning. A two-way ANOVA with repeated measures on translation parameters did not reveal significant main effects of TMS or time, or their interaction (all p 's > 0.05). A two-way ANOVA with repeated measures on rotation parameters revealed a significant main effect of time ($p = 0.01$), but no main effect of TMS and no time by TMS interaction (all p 's > 0.05). Additional analyses show that effects of TMS on global connectedness in OFC and LPFC (**Fig 2**) remain significant when correcting for head motion parameters.

Minor Concerns

1) Many of the statistics reported in the Results report “V” as a test statistic. I’m not familiar with this statistic. Perhaps a reference or a more conventional test statistic could be used.

We thank the reviewer for raising this point. The test statistic “V” refers to the Wilcoxon signed rank test (nonparametric version of the paired t-tests). We used this nonparametric test because it does not assume that dependent variables follow Gaussian distributions. There is no widely accepted notation for this test statistic compared to the well-known student T-statistic. We used “V” only because it is used by the R software package as default. Others have also used W or T to denote this statistic. We added a reference to this test when first mentioning it in the Method section (line 884-886).

We performed Wilcoxon signed-rank tests (“V”) for each ROI in each time bin to determine whether cTBS decreased global connectedness relative to sham (Hollander et al., 2013).

2) The authors compare the trial prior to reversals to trials following reversals in some analyses. It is unclear why this is an appropriate metric. Their computational model indicates that the trial following reversals is the critical locus of effect, so statistics that focus on that trial exclusively would seem to be more appropriate.

The reviewer is correct that the trial following reversals was the critical time point. However, only comparing a single time point across two sessions does not tell us whether the effects are specific for this trial or simply part of an overall session effect that can be observed in other trials as well. For instance, the behavioral prediction accuracy is expected to be lower specifically on the trial after relative to the trial before the reversal, and not just in general. This is even more important for fMRI responses, which are relative signals that can vary non-specifically across sessions. To rule out the possibility that effects are driven by some unspecific activity changes across sessions, comparisons between sessions are ideally performed on within-session differences. For instance, comparing responses to outcomes between reversal and trials after the reversal, ensures that effects of cTBS on the trial after reversal are driven by a slower return to baseline (as predicted by the model), rather than a general decrease in outcome-related activity. Finally, for the multi-voxel analysis for reward identity expectations, the trial prior to reversals served as a baseline to measure identity expectations. Specifically, the pattern similarity between the pre-reversal and reversal trial comes from trials with the same identity expectation, while the pattern similarity between the reversal and post-reversal trial comes from trials with different identity expectations. A positive difference between the pre-reversal similarity and post-reversal similarity is thus a measure of neural identity expectations. We extended the description for these analyses in the Result section (Lines 379-382):

This way, the trial prior to reversals serves as a baseline to measure identity expectations, analogous to using the pre-reversal trials as a baseline to measure behavioral adjustment from pre to post-reversals.

3) It has been documented that damage to the LPFC produces perseveration tendencies despite negative feedback (e.g. Wisconsin Card Sorting – Milner, 1963). Therefore, LPFC-cTBS may lead to impairments in reversal learning due to direct disruption of the LPFC in addition to, or in lieu of, disruption of the OFC. I believe a straightforward test of this would be to examine LPFC activation at the time of reversal feedback. Diminished LPFC activation following cTBS would be expected if LPFC-cTBS disrupts control processes that enable switching from irrelevant associations to new ones. Absence of such an effect would offer modest credence towards the idea that the OFC, not the LPFC is what is critical here.

We thank the reviewer for raising this point and suggesting the analysis. To directly address this point, we compared fMRI responses to reversal feedback in the LPFC stimulation site between the cTBS and sham session. This difference was not significant ($t(30) = 0.99, p = 0.17$). However, as pointed out in response to minor comment 2, we do not believe that this null result is very conclusive as this analysis does not involve a within-session control and therefore did not include it in the revised manuscript.

4) It appears that some analyses focus on the period of disruption identified by global connectedness (i.e. first run) and others consider all of the data. It seems that all analyses should be brought into alignment. Either effects are restricted to the first run or they are distribution across the entire scanning session. Waffling between these approaches gives the sense of a lack of rigor.

We agree with the reviewer that it would be ideal to consistently focus on data from the same time points. We initially designed the study and event-related fMRI analysis approaches to utilize data from all runs. We planned the global connectedness analyses to allow us to track the effects of our OFC network-targeted TMS approach over time. We felt this would be very important given that most of what is known about the longevity of cTBS effects comes from studies using cTBS on M1 and measuring MEPs (Wischnewski & Schutter, 2015). We frankly did not expect the effects to wear off that quickly. This forced us to reconsider our approach. While it is possible to address questions related to behavioral effects with less data, fMRI data are much noisier and thus fMRI analyses require more data to yield reliable results. We therefore permitted this inconsistency and hope the

reviewer agrees with our reasoning. We added the following change to lines 312-313 in the main manuscript to reflect our thoughts on this question:

To increase statistical power, this and the following fMRI analyses used data from all three runs.

5) In the methods, the univariate analyses make mention of cue-locked regressors, but I presume prediction error related effects are odor-locked. These regressors and contrasts upon them should be explicitly mentioned. If prediction errors are calculated on cue-locked regressors, a clear rationale for how this can map onto prediction errors would need to be established.

We thank the reviewer for pointing out that our description in the methods was not clear. Prediction error related effects are indeed based on odor-locked regressors. Cue-locked regressors were also included in these GLMs to account/control for any cue-induced neural activity. The PE contrasts were computed using beta estimates associated with the odor outcome regressors. We rewrote our description in the methods to clarify this point (lines 921-924).

We computed single-subject contrast images, comparing fMRI responses associated with the onset of odor delivery on reversal trials with responses on non-reversal trials across both sessions (*rev* vs. [*rev*₋₁, *rev*₊₁, *rev*₊₂]).

6) It appears that multi-variate analyses used an uncorrected threshold. This is not statistically permissible, certainly not in this age of heightened concern about reproducibility resulting from loose standards of the past.

We completely agree with the importance of applying appropriate statistical correction for voxel-wise hypothesis testing. However, in the current case, the purpose of the voxel-wise MVPA was to generate unbiased (w.r.t. cTBS effects) ROIs that we could use to test the effects of cTBS on neural representations of expected reward identity in an unbiased manner. To this end, we combined fMRI data from both sessions. Because we expected that identity expectations would be disrupted in half of the data (i.e., trials from cTBS sessions), we decided to use a more lenient threshold ($p = 0.005$, $k = 25$) to identify voxels that represented identity expectations and that were included in the unbiased ROI.

However, the reviewer is correct that we should not use the same lenient threshold to argue that there are identity expectations in the lateral OFC. To show evidence for these expectations, we now include a separate analysis using data from only the sham session. We tested for neural representations of expected reward identity using a threshold $p < 0.05$, SVC FWE corrected in the lateral OFC seed region. This revealed significant effects ($[-24, 44, -16]$, $t = 3.3$, $p_{\text{FWE}} = 0.045$ and $[-24, 36, -14]$, $t = 3.27$, $p_{\text{FWE}} = 0.049$). We now only reference the results from the corrected analysis to make claims about identity expectations in OFC but continue to use the unbiased ROI to assess effects of cTBS on OFC identity expectations. We included the additional analysis result in the legend of **Fig 6c** (line 406-409):

In a separate analysis using only data from the sham session, representations of expected reward identity in the left lateral OFC ($[-24, 44, -16]$, $t = 3.3$, $p_{\text{SVC-FWE}} = 0.045$; $[-24, 36, -14]$, $t = 3.27$, $p_{\text{SVC-FWE}} = 0.049$) survived correction for multiple comparisons within the lateral OFC seed region.

We also explain our reasoning for selecting this threshold on the results (line 386-389):

To define an unbiased ROI for comparing identity expectations between cTBS and sham, we performed a searchlight analysis to identify brain regions that represented reward identity expectations across both sham and cTBS sessions.

In contrast, we did not find significant effects within the LPFC stimulation site when testing for neural representations of expected reward identity in the sham session. We, therefore, removed all results and discussion related to identity expectations in the LPFC from the revised manuscript.

References

- Boorman, E. D., Rajendran, V. G., O'Reilly, J. X., & Behrens, T. E. (2016). Two anatomically and computationally distinct learning signals predict changes to stimulus-outcome associations in hippocampus. *Neuron*, *89*(6), 1343-1354.
- Delamater, A. R. (2007). The Role of the Orbitofrontal Cortex in Sensory-Specific Encoding of Associations in Pavlovian and Instrumental Conditioning. *Annals of the New York Academy of Sciences*, *1121*(1), 152-173. <https://doi.org/10.1196/annals.1401.030>
- Faget, L., Osakada, F., Duan, J., Ressler, R., Johnson, A. B., Proudfoot, J. A., Yoo, J. H., Callaway, E. M., & Hnasko, T. S. (2016). Afferent Inputs to Neurotransmitter-Defined Cell Types in the Ventral Tegmental Area. *Cell Rep*, *15*(12), 2796-2808. <https://doi.org/10.1016/j.celrep.2016.05.057>
- Geisler, S., Derst, C., Veh, R. W., & Zahm, D. S. (2007). Glutamatergic afferents of the ventral tegmental area in the rat. *J Neurosci*, *27*(21), 5730-5743. <https://doi.org/10.1523/JNEUROSCI.0012-07.2007>
- Hollander, M., Wolfe, D. A., & Chicken, E. (2013). *Nonparametric statistical methods*. John Wiley & Sons.
- Howard, J. D., Gottfried, J. A., Tobler, P. N., & Kahnt, T. (2015). Identity-specific coding of future rewards in the human orbitofrontal cortex. *Proceedings of the National Academy of Sciences*, *112*(16), 5195-5200. <https://doi.org/10.1073/pnas.1503550112>
- Howard, J. D., & Kahnt, T. (2017). Identity-Specific Reward Representations in Orbitofrontal Cortex Are Modulated by Selective Devaluation. *The Journal of Neuroscience*, *37*(10), 2627-2638. <https://doi.org/10.1523/JNEUROSCI.3473-16.2017>
- Howard, J. D., & Kahnt, T. (2018). Identity prediction errors in the human midbrain update reward-identity expectations in the orbitofrontal cortex. *Nature Communications*, *9*(1), 1611. <https://doi.org/10.1038/s41467-018-04055-5>
- Howard, J. D., Reynolds, R., Smith, D. E., Voss, J. L., Schoenbaum, G., & Kahnt, T. (2020). Targeted Stimulation of Human Orbitofrontal Networks Disrupts Outcome-Guided Behavior. *Current Biology*, *30*(3), 490-498.e494. <https://doi.org/10.1016/j.cub.2019.12.007>
- Howard, J. D., Reynolds, R., Smith, D. E., Voss, J. L., Schoenbaum, G., & Kahnt, T. (2020). Targeted Stimulation of Human Orbitofrontal Networks Disrupts Outcome-Guided Behavior. *Curr Biol*, *30*(3), 490-498 e494. <https://doi.org/10.1016/j.cub.2019.12.007>
- Klein-Flugge, M. C., Barron, H. C., Brodersen, K. H., Dolan, R. J., & Behrens, T. E. (2013). Segregated encoding of reward-identity and stimulus-reward associations in human orbitofrontal cortex. *J Neurosci*, *33*(7), 3202-3211. <https://doi.org/10.1523/JNEUROSCI.2532-12.2013>
- Logothetis, N. K., Pauls, J., Augath, M., Trinath, T., & Oeltermann, A. (2001). Neurophysiological investigation of the basis of the fMRI signal. *Nature*, *412*(6843), 150-157. <https://doi.org/10.1038/35084005>
- Mizrak, E., Bouffard, N. R., Libby, L. A., Boorman, E., & Ranganath, C. (2019). Representation of Task Structure in Human Hippocampus and Orbitofrontal Cortex. *bioRxiv*, 794305.
- Murty, V. P., Shermohammed, M., Smith, D. V., Carter, R. M., Huettel, S. A., & Adcock, R. A. (2014). Resting state networks distinguish human ventral tegmental area from substantia nigra. *NeuroImage*, *100*, 580-589. <https://doi.org/10.1016/j.neuroimage.2014.06.047>
- Pauli, W. M., Gentile, G., Collette, S., Tyszka, J. M., & O'Doherty, J. P. (2019). Evidence for model-based encoding of Pavlovian contingencies in the human brain. *Nat Commun*, *10*(1), 1099. <https://doi.org/10.1038/s41467-019-08922-7>
- Rudebeck, P. H., & Izquierdo, A. (2022). Foraging with the frontal cortex: A cross-species evaluation of reward-guided behavior. *Neuropsychopharmacology*, *47*(1), 134-146. <https://doi.org/10.1038/s41386-021-01140-0>
- Rudebeck, Peter H., & Murray, Elisabeth A. (2014). The Orbitofrontal Oracle: Cortical Mechanisms for the Prediction and Evaluation of Specific Behavioral Outcomes. *Neuron*, *84*(6), 1143-1156. <https://doi.org/10.1016/j.neuron.2014.10.049>
- Stalnaker, T. A., Cooch, N. K., McDannald, M. A., Liu, T.-L., Wied, H., & Schoenbaum, G. (2014). Orbitofrontal neurons infer the value and identity of predicted outcomes. *Nature Communications*, *5*(1), 3926. <https://doi.org/10.1038/ncomms4926>
- Stalnaker, T. A., Liu, T. L., Takahashi, Y. K., & Schoenbaum, G. (2018). Orbitofrontal neurons signal reward predictions, not reward prediction errors. *Neurobiol Learn Mem*, *153*(Pt B), 137-143. <https://doi.org/10.1016/j.nlm.2018.01.013>
- Suarez, J. A., Howard, J. D., Schoenbaum, G., & Kahnt, T. (2019). Sensory prediction errors in the human midbrain signal identity violations independent of perceptual distance. *eLife*, *8*, e43962. <https://doi.org/10.7554/eLife.43962>

- Tomov, M. S., Dorfman, H. M., & Gershman, S. J. (2018). Neural computations underlying causal structure learning. *Journal of Neuroscience*, 38(32), 7143-7157.
- Tomov, M. S., Tsvividis, P. A., Pouncy, T., Tenenbaum, J. B., & Gershman, S. J. (2023). The neural architecture of theory-based reinforcement learning. *Neuron*. <https://doi.org/10.1016/j.neuron.2023.01.023>
- Vaidya, A. R., Jones, H. M., Castillo, J., & Badre, D. (2021). Neural representation of abstract task structure during generalization. *eLife*, 10, e63226.
- Wang, F., Howard, J. D., Voss, J. L., Schoenbaum, G., & Kahnt, T. (2020). Targeted Stimulation of an Orbitofrontal Network Disrupts Decisions Based on Inferred, Not Experienced Outcomes. *J Neurosci*, 40(45), 8726-8733. <https://doi.org/10.1523/JNEUROSCI.1680-20.2020>
- Watabe-Uchida, M., Eshel, N., & Uchida, N. (2017). Neural Circuitry of Reward Prediction Error. *Annu Rev Neurosci*, 40, 373-394. <https://doi.org/10.1146/annurev-neuro-072116-031109>
- Watabe-Uchida, M., Zhu, L., Ogawa, S. K., Vamanrao, A., & Uchida, N. (2012). Whole-brain mapping of direct inputs to midbrain dopamine neurons. *Neuron*, 74(5), 858-873. <https://doi.org/10.1016/j.neuron.2012.03.017>
- Wickersham, I. R., Lyon, D. C., Barnard, R. J., Mori, T., Finke, S., Conzelmann, K. K., Young, J. A., & Callaway, E. M. (2007). Monosynaptic restriction of transsynaptic tracing from single, genetically targeted neurons. *Neuron*, 53(5), 639-647. <https://doi.org/10.1016/j.neuron.2007.01.033>
- Wischniewski, M., & Schutter, D. J. (2015). Efficacy and Time Course of Theta Burst Stimulation in Healthy Humans. *Brain Stimul*, 8(4), 685-692. <https://doi.org/10.1016/j.brs.2015.03.004>

REVIEWERS' COMMENTS

Reviewer #1 (Remarks to the Author):

The authors have done a very thorough job of responding to my previous review. I was particularly impressed by the specificity of the effects to lateral OFC. I have no further comments or concerns.

Reviewer #2 (Remarks to the Author):

Thank you for your comprehensive and thoughtful response to my comments. The significant revisions and additional analyses you conducted provide robust support for the role of the lateral OFC in regulating identity prediction errors. This addition, along with the enhanced clarity in the methods section, substantially improves the manuscript's overall quality. I also appreciate the careful incorporation and discussion of the additional RSA. Your efforts have successfully addressed all my concerns. I eagerly anticipate the publication of your final work.

Reviewer #3 (Remarks to the Author):

In my previous review, I raised various concerns regarding potential confounds and analytic choices. The authors have been mostly responsive to my concerns. My remaining concerns are:

1) Analyses ruling out physical discomfort and perceived intensity as confounds should be added formally in Extended Data. At the present, these are mentioned in only a cursory manner in the Methods.

2) I appreciate the analyses that rule out motion as a confound. In addition to the analyses presented, I would suggest that the authors repeat the analyses using framewise displacement (e.g. Power et al., 2012), which is a more conventional motion metric that combines the rotation and translation parameters into a single quantity.

REVIEWERS' COMMENTS

Reviewer #1 (Remarks to the Author):

The authors have done a very thorough job of responding to my previous review. I was particularly impressed by the specificity of the effects to lateral OFC. I have no further comments or concerns.

We thank for the reviewer's positive feedback. We are glad to hear that we have been able to address all comments and believe they have significantly contributed to shaping the manuscript.

Reviewer #2 (Remarks to the Author):

Thank you for your comprehensive and thoughtful response to my comments. The significant revisions and additional analyses you conducted provide robust support for the role of the lateral OFC in regulating identity prediction errors. This addition, along with the enhanced clarity in the methods section, substantially improves the manuscript's overall quality. I also appreciate the careful incorporation and discussion of the additional RSA.

Your efforts have successfully addressed all my concerns. I eagerly anticipate the publication of your final work.

We thank for the reviewer's positive evaluation. We are happy to hear that we have comprehensively addressed all concerns. We feel these revisions have improved the manuscript significantly.

Reviewer #3 (Remarks to the Author):

In my previous review, I raised various concerns regarding potential confounds and analytic choices. The authors have been mostly responsive to my concerns. My remaining concerns are:

We are glad to hear that we have been able to address most of the initial concerns. Below, we address the remaining two comments.

1) Analyses ruling out physical discomfort and perceived intensity as confounds should be added formally in Extended Data. At the present, these are mentioned in only a cursory manner in the Methods.

We thank the reviewer for this suggestion. These analyses and results are now described in more details in the Supplementary Information.

2) I appreciate the analyses that rule out motion as a confound. In addition to the analyses presented, I would suggest that the authors repeat the analyses using framewise displacement (e.g. Power et al., 2012), which is a more conventional motion metric that combines the rotation and translation parameters into a single quantity.

We thank the reviewer for suggesting using framewise displacement (FD), the conventional motion metric. We now run additional analyses using FD. Specifically, we calculated the instantaneous FD values (Power et al., 2012) for each volume and then calculated the sum of FD within each half of every run to bring them on the same time scale as the global connectedness analysis.

In the first step, similar to the analyses conducted for translation and rotation parameters, we conducted a two-way repeated measure ANOVA to examine the effect of TMS and time on FD. This analysis did not reveal any significant main effect of TMS or time, or their interaction (all p 's > 0.05). We further tested whether the effect of TMS on the global connectedness in the lateral OFC and LPFC persists when accounting for the effects of head motion. For this analysis, we repeated the comparison of two linear mixed effect models: a full model with effects from subjects, TMS session, FD as predictors, and a reduced model without the TMS session effect. This analysis was conducted once for the OFC (focused on time bins 1 and 2) and once for the LPFC (focused on time bins 2, 3, and 4). Similar to our previous model comparison results from using translation and rotation parameters, this new analysis revealed that adding TMS session as a predictor significantly improved the model fit for both the OFC (time bin 1: $p = 0.00027$; time bin 2: $p = 0.00066$) and LPFC (time bin 2: $p = 7.455e-5$; time bin 3: 0.014; time bin 4: $p = 8.62e-5$), demonstrating that the effect of TMS on global connectedness was not driven by differences in head motion.

We have now updated our **Supplementary Fig 4** to reflect the new analysis.

Supplementary Fig 4: Comparison of head motion parameters across runs between cTBS and sham sessions. (a) Translation parameters, (b) rotation parameters, and (c) framewise displacement during scanning. A two-way ANOVA with repeated measures on Translation parameters or framewise displacement did not reveal significant main effects of TMS or time, or their interaction (all p 's > 0.05). A two-way ANOVA with repeated measures on rotation parameters revealed a significant main effect of time ($p = 0.01$), but no main effect of TMS and no time by TMS interaction (all p 's > 0.05). Additional analyses show that effects of TMS on global connectedness in OFC and LPFC (**Fig 2**) remain significant when correcting for these head motion parameters. For **a, b, c**, $n = 31$.

Power, J. D., Barnes, K. A., Snyder, A. Z., Schlaggar, B. L., & Petersen, S. E. (2012). Spurious but systematic correlations in functional connectivity MRI networks arise from subject motion. *NeuroImage*, 59(3), 2142-2154. <https://doi.org/10.1016/j.neuroimage.2011.10.018>